# Effective Exploration Based on the Structural Information Principles

**Xianghua Zeng**[1], **Hao Peng**[1], **Angsheng Li**[1,2]

[1] State Key Laboratory of Software Development Environment, Beihang University, Beijing, China
[2] Zhongguancun Laboratory, Beijing, China
{zengxianghua, penghao, angsheng}@buaa.edu.cn, liangsheng@gmail.zgclab.edu.cn

## Abstract

Traditional information theory provides a valuable foundation for Reinforcement Learning (RL), particularly through representation learning and entropy maximization for agent exploration. However, existing methods primarily concentrate on modeling the uncertainty associated with RL's random variables, neglecting the inherent structure within the state and action spaces. In this paper, we propose a novel **S**tructural **I**nformation principles-based **E**ffective **E**xploration framework, namely **SI2E**. Structural mutual information between two variables is defined to address the single-variable limitation in structural information, and an innovative embedding principle is presented to capture dynamics-relevant state-action representations. The SI2E analyzes value differences in the agent's policy between state-action pairs and minimizes structural entropy to derive the hierarchical state-action structure, referred to as the encoding tree. Under this tree structure, value-conditional structural entropy is defined and maximized to design an intrinsic reward mechanism that avoids redundant transitions and promotes enhanced coverage in the state-action space. Theoretical connections are established between SI2E and classical information-theoretic methodologies, highlighting our framework's rationality and advantage. Comprehensive evaluations in the MiniGrid, MetaWorld, and DeepMind Control Suite benchmarks demonstrate that SI2E significantly outperforms state-of-the-art exploration baselines regarding final performance and sample efficiency, with maximum improvements of $37.63\%$ and $60.25\%$, respectively.

## 1 Introduction

Reinforcement Learning (RL) has emerged as a pivotal technique for addressing sequential decision-making problems, including game intelligence [Vinyals et al., 2019, Badia et al., 2020], robotic control [Andrychowicz et al., 2017, Liu and Abbeel, 2021], and autonomous driving [Prathiba et al., 2021, Pérez-Gil et al., 2022]. In the realm of RL, striking a balance between exploration and exploitation is crucial for optimizing agent policies and mitigating the risk of suboptimal outcomes, especially in scenarios characterized by high dimensions and sparse rewards [Zhang et al., 2021b].

Recently, advancements in information-theoretic approaches have shown promise for exploration in self-supervised settings. The maximum entropy framework over the action space [Haarnoja et al., 2017] has led to the development of robust algorithms such as Soft Q-learning [Nachum et al., 2017], SAC [Haarnoja et al., 2018], and MPO [Abdolmaleki et al., 2018]. Additionally, various objectives focused on maximizing state entropy are utilized to ensure comprehensive state coverage [Hazan et al., 2019, Islam et al., 2019]. To facilitate the exploration of complex state-action pairs, MaxRenyi optimizes Rényi entropy across the state-action space [Zhang et al., 2021a]. However, a prevalent issue with entropy maximization strategies is their tendency to bias exploration towards low-value states, making them vulnerable to imbalanced state-value distributions in supervised settings. To mitigate

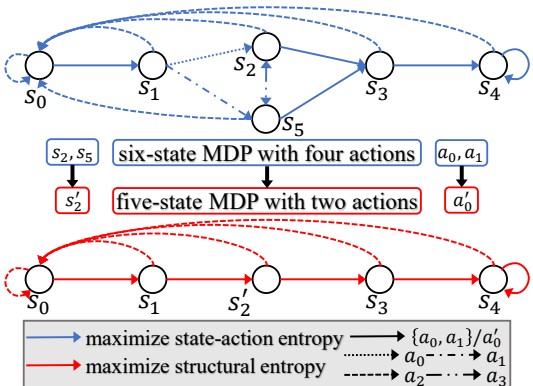

Figure 1: By incorporating the inherent state-action structure, we simplify the original six-state Markov Decision Process (MDP) with four actions to a five-state MDP with two actions, effectively reducing the size of state-action space from $24(6 \times 4)$ to $10(5 \times 2)$. Here, $s_2'$ and $a_0'$ represent vertex communities $\{s_2, s_5\}$ and $\{a_0, a_1\}$, respectively. In this scenario, a policy maximizing state-action Shannon entropy would encompass all possible transitions (blue color). In contrast, a policy maximizing structural entropy would selectively focus on crucial transitions (red color), avoiding redundant transitions between $s_2$ and $s_5$.

this, value-conditional state entropy is introduced to compute intrinsic rewards based on the estimated values of visited states [Kim et al., 2023]. Due to their instability in noisy and high-dimensional environments, a Dynamic Bottleneck (DB) [Bai et al., 2021] is developed based on the Information Bottleneck (IB) principle [Tishby et al., 2000], thereby obtaining dynamics-relevant representations of state-action pairs. Despite their successes, existing information-theoretic exploration methods have a critical limitation: they often overlook the inherent structure within state and action spaces. This oversight necessitates new approaches to enhance exploration effectiveness.

Figure 1 illustrates a simple six-state Markov Decision Process (MDP) with four actions. The different densities of the blue and red lines represent different actions, as indicated in the legend, leading to state transitions aimed at optimizing the return to the initial state $s_0$. Solid lines specifically denote actions $a_0$ and $a_1$. The transitions between states $s_2$ and $s_5$ are deemed redundant as they do not facilitate the primary objective of efficiently returning to $s_0$. Therefore, the state-action pairs $(s_2, a_3)$ and $(s_5, a_3)$ have lower policy values. A policy maximizing state-action Shannon entropy would encompass all possible transitions (blue color). In contrast, a policy incorporating the inherent state-action structure will divide these redundant state-action pairs into a vertex sub-community and minimize the entropy of this sub-community to avoid visiting it unnecessarily. Simultaneously, it maximizes state-action entropy, resulting in maximal coverage for transitions (red color) that are more likely to contribute to the desired outcome in the simplified five-state MDP.

Departing from traditional information theory applied to random variables, structural information [Li and Pan, 2016] has been devised to quantify dynamic uncertainty within complex graphs under a hierarchical partitioning structure known as an "encoding tree". Structural entropy is conceptualized as the minimum number of bits required to encode a vertex accessible through a single-step random walk and is minimized to optimize the encoding tree. However, this definition is limited to single-variable graphs and cannot capture the structural relationship between two variables. While the underlying graph can encompass multiple variables, current structural information principles are limited in treating these variables as a single joint variable, measuring only its structural entropy. This limitation prevents the effective quantification of structural similarity between variables, inherently imposing a single-variable constraint. Prior research on reinforcement learning using structural information principles [Zeng et al., 2023b,c] has focused on independently modeling state or action variables without simultaneously considering state-action representations.

In this work, we propose SI2E, a novel and unified framework grounded in structural information principles for effective exploration within high-dimensional and sparse-reward environments. Initially, we embed state-action pairs into a low-dimensional space and present an innovative representation learning principle to capture dynamics-relevant information and compress dynamics-irrelevant information. Then, we strategically increase the state-action pairs' structural mutual information with subsequent states while decreasing it with current states. We analyze value differences among state-action representations to form a complete graph and minimize its structural entropy to derive the

optimal encoding tree, thereby unveiling the hierarchical community structure of state-action pairs. By leveraging this identified structure, we design an intrinsic reward mechanism tailored to avoid redundant transitions and enhance maximal coverage in exploring the state-action space. Furthermore, we establish theoretical connections between our framework and classical information-theoretic methodologies, highlighting the rationality and advantage of SI2E. Our thorough evaluations across diverse and challenging tasks in the MiniGrid, MetaWorld, and DeepMind Control Suite benchmarks have consistently shown SI2E's superiority, with significant improvements in final performance and sample efficiency, surpassing state-of-the-art exploration baselines. For further research, the source code is available at [1]. Our contributions are summarized as follows:

- A novel framework based on structural information principles, SI2E, is proposed for effective exploration in high-dimensional RL environments with sparse rewards.
- An innovative principle of structural mutual information is introduced to overcome the single-variable constraint inherent in existing structural information and to enhance the acquisition of dynamics-relevant representations for state-action pairs.
- A unique intrinsic reward mechanism that maximizes the value-conditional structural entropy is designed to avoid redundant transitions and promote enhanced coverage in the state-action space.
- Our experiments on various challenging tasks demonstrate that SI2E significantly improves final performance and sample efficiency by up to 37.63% and 60.25%, respectively, compared to state-of-the-art baselines.

## 2 Preliminaries

In this section, we formalize the definitions of fundamental concepts. The descriptions of primary notations are summarized in Appendix A.1 for ease of reference.

### 2.1 Traditional Information Principles

Consider the random variable pair $Z = (X, Y)$ with a joint distribution probability denoted by $p(x, y) \in (0, 1)$. The marginal probabilities, $p(x)$ and $p(y)$, are defined as $p(x) = \sum_y p(x, y)$ and $p(y) = \sum_x p(x, y)$, respectively. The joint Shannon entropy [Shannon, 1953] of $X$ and $Y$ is $H(X, Y) = -\sum_{(x,y)} [p(x, y) \cdot \log p(x, y)]$, which quantifies the total uncertainty in $Z$. Conversely, the marginal entropies $H(X) = -\sum_x [p(x) \cdot \log p(x)]$ and $H(Y) = -\sum_y [p(y) \cdot \log p(y)]$ characterize the uncertainty in $X$ and $Y$ individually. The mutual information $I(X; Y) = \sum_{x,y} \left[ p(x, y) \cdot \log \frac{p(x,y)}{p(x)p(y)} \right]$ quantifies the shared uncertainty between $X$ and $Y$. It satisfies the following relationship: $I(X; Y) = H(X) + H(Y) - H(X, Y)$.

### 2.2 Reinforcement Learning

Within the context of RL, the sequential decision-making problem is formalized as a Markov Decision Process (MDP) [Bellman, 1957]. The MDP is characterized by a tuple $(\mathcal{O}, \mathcal{A}, \mathcal{P}, \mathcal{R}^e, \gamma)$, where $\mathcal{O}$ denotes the observation space, $\mathcal{A}$ the action space, $\mathcal{P}$ the environmental transition function, $\mathcal{R}^e$ the extrinsic reward function, and $\gamma \in [0, 1)$ the discount factor. At each discrete timestep $t$, the agent selects an action $a_t \in \mathcal{A}$ upon observing $o_t \in \mathcal{O}$. This leads to a transition to a new observation $o_{t+1} \sim \mathcal{P}(o_t, a_t)$ and a reward $r_t^e \in \mathbb{R}$. The policy network $\pi$ is optimized to maximize the cumulative long-term expected discounted reward.

**Maximum State Entropy Exploration.** In environments with sparse rewards, agents are encouraged to explore the state space extensively, which can be incentivized by maximizing the Shannon entropy $H(S)$ of state variable $S$. When the prior distribution $p(s)$ is not available, the non-parametric $k$-nearest neighbors ($k$-NN) entropy estimator [Singh et al., 2003] is employed. For a given set of $n$ independent and identically distributed samples from a $d_x$-dimensional space $\{x_i\}_{i=0}^{n-1}$, the entropy of variable $X$ is estimated as follows:

$$\widehat{H}_{KL}(X) = \frac{d_x}{n} \sum_{i=0}^{n-1} \log d(x_i) + C, \tag{1}$$

---

[1] https://github.com/SELGroup/SI2E

where $d(x_i)$ is twice the distance from $x_i$ to its $k$-th nearest neighbor, and $C$ is a constant term.

**Information Bottleneck Principle.** In the supervised learning paradigm, representation learning aims to transform an input source $X$ into a representation $Z$, targeted towards an output source $Y$. The Information Bottleneck (IB) principle [Tishby et al., 2000] refines this process by maximizing the mutual information $I(Z; Y)$ between $Z$ and $Y$, capturing the relevant features of $Y$ within $Z$. Concurrently, the IB principle imposes a complexity constraint by minimizing the mutual information $I(Z; X)$ between $Z$ and $X$, effectively discarding irrelevant features. To balance these objectives, the IB principle utilizes a Lagrangian multiplier, facilitating a balanced trade-off between the richness of the representation and its complexity.

## 2.3 Structural Information Principles

The encoding tree $T$ of an undirected and weighted graph $G = (V, E)$ is characterized as a rooted tree with the following properties: 1) Each tree node $\alpha$ in $T$ corresponds to a subset of graph vertices $V_\alpha \subseteq V$. 2) The subset $V_\lambda$ of tree root $\lambda$ encompasses all vertices in $V$. 3) Each subset $V_\nu$ of a leaf node $\nu$ in $T$ only contains a single vertex $v$, thus $V_\nu = \{v\}$. 4) For each non-leaf node $\alpha$, the number of its children is assumed as $l_\alpha$, with the $i$-th child specified as $\alpha_i$. The collection of subsets $V_{\alpha_1}, \ldots, V_{\alpha_{l_\alpha}}$ constitutes a sub-partition of $V_\alpha$.

Given an encoding tree $T$ whose height is at most $K$, the $K$-dimensional structural entropy of graph $G$ is defined as follows:

$$H^T(G) = - \sum_{\alpha \in T, \alpha \neq \lambda} \left[ \frac{g_\alpha}{\text{vol}(G)} \cdot \log \frac{\text{vol}(\alpha)}{\text{vol}(\alpha^-)} \right], \quad H^K(G) = \min_T H^T(G), \qquad (2)$$

where $g_\alpha$ is the weighted sum of all edges connecting vertices within the subset $V_\alpha$ to vertices outside the subset $V_\alpha$.

# 3 Structural Mutual Information

In this section, we address the single-variable constraint prevalent in existing structural information principles and introduce the concept of structural mutual information for subsequent state-action representation learning within our SI2E framework.

Given the random variable pair $(X, Y)$ with $|X| = |Y| = n$, we construct an undirected bipartite graph $G_{xy}$ to represent the joint distribution of $X$ and $Y$. In $G_{xy}$, each vertex $x \in X$ connects to each vertex $y \in Y$ via weighted edges, where the weight of each edge equals the joint probability $p(x, y)$. Notably, no edges connect vertices within the same set, $X$ or $Y$, and the total sum of the edge weights is 1, $\sum_{x,y} p(x, y) = 1$. Each single-step random walk in $G_{xy}$ accesses either a vertex from $X$ or $Y$. The structural entropy of variable $X$ in $G_{xy}$ is defined as the number of bits required to encode all accessible vertices in the set $X$. It is calculated using the following formula:

$$H^{SI}(X) = - \sum_{x \in X} \left[ \frac{p(x)}{\text{vol}(G_{xy})} \cdot \log \frac{p(x)}{\text{vol}(G_{xy})} \right] = - \sum_{x \in X} \left[ \frac{p(x)}{2} \cdot \log \frac{p(x)}{2} \right], \qquad (3)$$

where the sum of all vertex degrees is twice the total sum of edge weights, resulting in $\text{vol}(G_{xy}) = 2$.

The structural entropy $H^{SI}(Y)$ is defined similarly. We restrict the partitioning structure of $G_{xy}$ to 2-layer approximate binary trees, denoted as $\mathcal{T}^2$, to calculate the required bits to encode accessible vertices in $X$ or $Y$, defined as the joint structural entropy. This tree structure mandates that each intermediate node (neither root nor leaf) has precisely two children. We begin by initializing a one-layer encoding tree, $T_{xy}^0$, designating each non-root node $\alpha$'s parent as the root $\lambda$, with $\alpha^- = \lambda$. By applying the stretch operator from the HCSE algorithm [Pan et al., 2021], we pursue an iterative and greedy optimization of $T_{xy}^0$, further detailed in Appendix A.2. The optimal encoding tree, $T_{xy}^*$, for $G_{xy}$ and the joint entropy under $T_{xy}^*$ are achieved through:

$$T_{xy}^* = \arg \min_{T \in \mathcal{T}^2} H^T(G_{xy}), \quad H^{T_{xy}^*}(X, Y) = H^{T_{xy}^*}(G_{xy}). \qquad (4)$$

Utilizing 2-layer approximate binary trees as the structural framework ensures computational tractability and more complex structures will increase the cost of increased computational complexity, which can be prohibitive for practical applications.

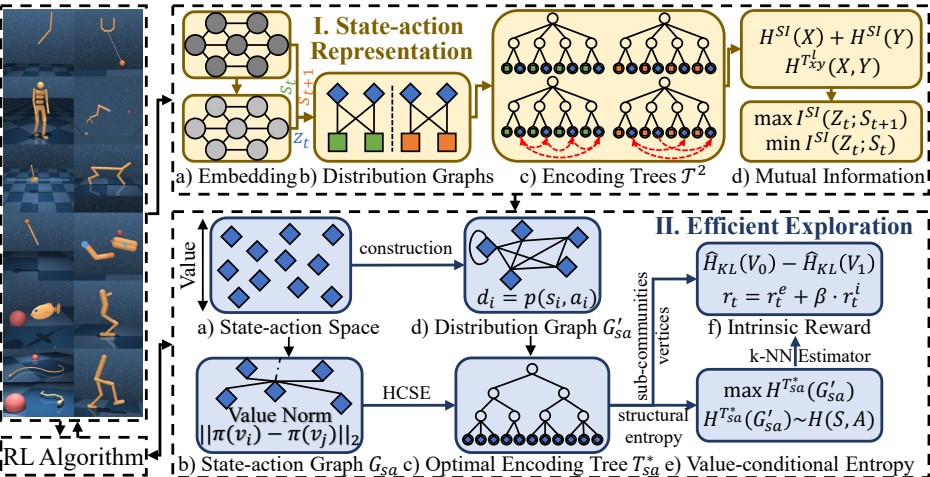

Figure 2: The SI2E's overview architecture, including state-action representation learning and maximum structural entropy exploration.

We derive the following proposition regarding $\mathcal{T}^2$, with the detailed proof provided in Appendix B.1.

**Proposition 3.1.** *Consider an undirected graph $G = (V, E)$ with vertices $v_i$ and $v_j$ in $V$. If the edge $(v_i, v_j)$ is absent from $E$, then in the 2-layer approximate binary optimal encoding tree $T^* \in \mathcal{T}^2$, there does not exist any non-root node $\alpha$ such that both $v_i$ and $v_j$ are included in its subset $V_\alpha$.*

Each intermediate node $\alpha \in T_{xy}^*$ corresponds to a subset comprising exactly one $x$ vertex and one $y$ vertex, thus establishing a one-to-one matching structure between variables $X$ and $Y$. The $i$-th intermediate node in $T_{xy}^*$, ordered from left to right, is denoted as $\alpha_i$. Within this subset, the $x$ and $y$ vertices are labeled as $x_i$ and $y_i$, respectively.

To define structural mutual information accurately, it is essential to consider the joint entropy of two variables under various partition structures. We introduce an $l$-transformation applied to $T_{xy}^*$ to systematically traverse all potential one-to-one matching of these variables, providing a comprehensive measure of their structural similarity. Given an integer parameter $l > 0$, this transformation generates a new 2-layer approximate binary tree, $T_{xy}^l$, representing an alternative one-to-one matching structure.

**Definition 3.2.** *For each intermediate node $\alpha_i$ in $T_{xy}^l$, the $x$ and $y$ vertices in $T_{\alpha_i}^l$ are specified as $T_{\alpha_i}^l = \{x_{i'}, y_i\}$, where $i' = (i + l) \bmod n$.*

The resulting tree $T_{xy}^l$ is equivalent to the optimal tree $T_{xy}^*$ when $l = 0$. We provide an example in Appendix A.3 with $n = 4$ for an intuitive understanding of the described process.

**Definition 3.3.** *Leveraging the relationship $I(X; Y) = H(X) + H(Y) - H(X, Y)$ in traditional information theory, we formally define the structural mutual information, $I^{SI}(X; Y)$, as follows:*

$$I^{SI}(X; Y) = \sum_{l=0}^{n-1} \left[ H^{SI}(X) + H^{SI}(Y) - H^{T_{xy}^l}(X, Y) \right] = \sum_{i,j} \left[ p(x_i, y_j) \cdot \log \frac{2}{p(x_i) + p(y_j)} \right]. \tag{5}$$

The detailed derivation is provided in Appendix C.1. Structural mutual information quantifies the average difference between the required encoding bits of accessible vertices of a single variable and the joint variable. The following theorem outlines the connection between $I^{SI}(X; Y)$ and $I(X; Y)$, with a detailed proof in Appendix B.2.

**Theorem 3.4.** *For a tuning parameter $0 \leq \epsilon \leq 1$, its holds for the structural mutual information $I^{SI}(X; Y)$, traditional mutual information $I(X; Y)$, and joint Shannon entropy $H(X, Y)$ that:*

$$I(X; Y) \leq I^{SI}(X; Y) \leq I(X; Y) + (1 - \epsilon) \cdot H(X, Y). \tag{6}$$

## 4 The Proposed SI2E Framework

In this section, we describe the detailed designs of the proposed SI2E framework, which captures dynamic-relevant state-action representations through structural mutual information (see Section

4.1) and enhances state-action coverage conditioned by the agent's policy by maximizing structural entropy (see Section 4.2). The overall architecture of our framework is illustrated in Figure 2.

## 4.1 State-action Representation Learning

To effectively learn dynamics-relevant state-action representations, we present an innovative embedding principle that maximizes the structural mutual information with subsequent states and minimizes it with current states.

**Structural Mutual Information Principle.** In this phase, the input variables at timestep $t$ encompass the current observation $O_t$ and the action $A_t$, with the target being the subsequent observation $O_{t+1}$. We denote the encoding of observations $O_t$ and $O_{t+1}$ as states $S_t$ and $S_{t+1}$, respectively. We aim to generate a latent representation $Z_t$ for the tuple $(S_t, A_t)$, which preserves information relevant to $S_{t+1}$ while compressing information pertinent to $S_t$. This embedding process mentioned above is detailed as follows:

$$S_t = f_s(O_t), \quad S_{t+1} = f_s(O_{t+1}), \quad Z_t = f_z(S_t, A_t), \tag{7}$$

where $f_s$ and $f_z$ are the respective encoders for states and state-action pairs (step I. a in Figure 2). For the state-action embeddings $Z_t$, we construct two undirected bipartite graphs, $G_{zs}$ and $G_{zs'}$, as shown in step I. b of Figure 2. These graphs represent the joint distributions of $Z_t$ with the current states $S_t$ and subsequent states $S_{t+1}$. In step I. c of Figure 2, we generate 2-layer approximate binary trees for $G_{zs}$ and $G_{zs'}$ and calculate the mutual information $I^{SI}(Z_t; S_t)$ and $I^{SI}(Z_t; S_{t+1})$ using Equation 5. Building upon the Information Bottleneck (IB) [Tishby et al., 2000], we present an embedding principle that aims to minimize $I^{SI}(Z_t; S_t)$ while maximizing $I^{SI}(Z_t; S_{t+1})$, as demonstrated in step I. d of Figure 2. When the joint distribution between variables $Z_t$ and $S_{t+1}$ shows a one-to-one correspondence-meaning for each $z_t \in Z_t$ value, there is a unique $s_{t+1} \in S_{t+1}$ corresponding to it, and vice versa-their mutual information takes its maximum value. We introduce a theorem to elucidate the equivalence between $I^{SI}(Z_t; S_{t+1})$ and $I(Z_t; S_{t+1})$ under this condition.

**Theorem 4.1.** *For a joint distribution of variables $X$ and $Y$ that shows a one-to-one correspondence, $I^{SI}(X; Y)$ equals $I(X; Y)$.*

A detailed proof is provided in Appendix B.3. When $Z_t$ and $S_t$ are mutually independent, the mutual information $I(Z_t; S_t)$ attains its minimum value. Our $I^{SI}(Z_t; S_t)$ goes beyond this, incorporating the joint entropy $H(Z_t, S_t)$ according to Theorem 3.4. This integration effectively eliminates the irrelevant information embedded in the representation variable $Z_t$, a significant step in our research. Consequently, structural mutual information can be considered a reasonable and desirable learning objective for acquiring dynamics-relevant state-action representations.

**Representation Learning Objective.** Due to the computational challenges of directly minimizing $I^{SI}(Z_t; S_t)$, we formulate a variational upper bound $I(Z_t; S_t) + H(Z_t|S_t) + H(S_t)$ (see Appendix C.2). Noting that the term $H(S_t)$ is extraneous to our model, we equate the minimization of $I^{SI}(Z_t; S_t)$ to the minimization of $I(Z_t; S_t)$ and $H(Z_t|S_t)$.

By employing a feasible decoder to approximate the marginal distribution of $Z_t$, we derive an upper bound of $I(Z_t; S_t)$ (See Appendix C.3) as follows:

$$I(Z_t; S_t) \leq \sum \left[ p(z_t, s_t) \cdot D_{KL}(p(z_t|s_t)||q_m(z_t)) \right] \triangleq L_{up}. \tag{8}$$

To concurrently decrease the conditional entropy $H(Z_t|S_t)$, we introduce a predictive objective (See Appendix C.4) through a tractable decoder $q_{z|s}$ for the conditional probability $p(z_t|s_t)$ as follows:

$$H(Z_t|S_t) \leq \sum \left[ p(z_t, s_t) \cdot \log \frac{1}{q_{z|s}(z_t|s_t)} \right] \triangleq L_{z|s}, \tag{9}$$

where $L_{z|s}$ represents the log-likelihood of $Z_t$ given $S_t$.

To efficiently optimize $I^{SI}(Z_t; S_{t+1})$, we maximize its lower bound, $I(Z_t; S_{t+1})$, as detailed in Theorem 3.4. By utilizing an alternative decoder $q_{s|z}$ for the conditional probability $p(s_{t+1}|z_t)$, we obtain a lower bound of $I(Z_t; S_{t+1})$ (See Appendix C.5) as follows:

$$I(Z_t; S_{t+1}) \geq \sum \left[ p(z_t, s_{t+1}) \cdot \log q_{s|z}(s_{t+1}|z_t) \right] \triangleq L_{s|z}, \tag{10}$$

where $L_{s|z}$ denotes the log-likelihood of $S_{t+1}$ conditioned on $Z_t$.

Within our SI2E framework, the definitive loss for representation learning is a combination of the above bounds, $L = L_{up} + L_{z|s} + \eta \cdot L_{s|z}$, where $\eta$ is a Lagrange multiplier used to maintain equilibrium among the specified terms.

## 4.2 Maximum Structural Entropy Exploration

We have designed a unique intrinsic reward mechanism to address the challenge of imbalance exploration towards low-value states in traditional entropy strategies, as discussed by [Kim et al., 2023]. Specifically, we generate a hierarchical state-action structure based on the agent's policy and define value-conditional structural entropy as an intrinsic reward for effective exploration.

**Hierarchical State-action Structure.** Derived from the history of agent-environment interactions, we extract state-action pairs (step II. a in Figure 2) to form a complete graph $G_{sa}$ (step II. b in Figure 2) that encapsulates the value relationships caused by the agent's policy. Within this graph, any two vertices $v_i$ and $v_j$ is connected by an undirected edge whose weight $w_{ij}$ is determined as: $w_{ij} = ||\pi(s_t^i, a_t^i) - \pi(s_t^j, a_t^j)||_2$. The state-action pairs $(s_t^i, a_t^i)$ and $(s_t^j, a_t^j)$ are associated with vertices $v_i$ and $v_j$, respectively. We minimize the 2-dimensional structural entropy of this graph $G_{sa}$ to generate its 2-layer optimal encoding tree, denoted as $T_{sa}^*$ (step II. c in Figure 2). This tree $T_{sa}^*$ delineates a hierarchical community structure among the state-action vertices, with the root node corresponding to a community encompassing all vertices. Each intermediate node in $T_{sa}^*$ corresponds to a sub-community, including vertices that share similar $\pi$ values.

**Value-conditional Structural Entropy.** To measure the extent of the policy's coverage across the state-action space, we construct an additional distribution graph $G'_{sa}$ (step II. d in Figure 2). The graph $G'_{sa}$ shares the same vertex set as $G_{sa}$. The following proposition confirms the existence of such a graph, with a detailed proof provided in Appendix B.4.

**Proposition 4.2.** *Given positive visitation probabilities $p(s_t^0, a_t^0), \ldots, p(s_t^{n-1}, a_t^{n-1})$ for all state-action pairs, there exists a weighted, undirected, and connected graph $G'_{sa}$, where each vertex's degree $d_i$ equals its visitation probability $p(s_t^i, a_t^i)$.*

In the graph $G'_{sa}$, the set of all state-action vertices is denoted as $V_0$, and the set of all state-action sub-communities is denoted as $V_1$. The Shannon entropies associated with the distribution of visitation probabilities for these sets are represented as $H(V_0)$ and $H(V_1)$, respectively, where $H(V_0) = H(S_t, A_t)$. Within the 2-layer state-action community represented by $T_{sa}^*$, we define the structural entropy of $G'_{sa}$ using Equation 2, denoted as $H^{T_{sa}^*}(G'_{sa})$ (step II. e in Figure 2). The following theorem delineates the relationship between the value-conditional entropy $H^{T_{sa}^*}(G'_{sa})$ with the state-action Shannon entropy $H(S_t, A_t)$. A detailed proof is provided in Appendix B.5.

**Theorem 4.3.** *For a tuning parameter $0 \leq \zeta \leq 1$, it holds for the structural entropy $H^{T_{sa}^*}(G'_{sa})$ and the Shannon entropy $H(S_t, A_t)$ that:*

$$\zeta \cdot H(S_t, A_t) \leq H(V_0) - H(V_1) \leq H^{T_{sa}^*}(G'_{sa}) \leq H(S_t, A_t), \tag{11}$$

where $H(V_0) - H(V_1)$ is a variational lower bound of $H^{T_{sa}^*}(G'_{sa})$. On the one hand, the term $H(V_0)$ ensures maximal coverage of the entire state-action space, analogous to the traditional Shannon entropy. On the other hand, the term $H(V_1)$ mitigates uniform coverage among state-action sub-communities with diverse $\pi$ values, thus addressing the challenge of imbalance exploration. By identifying the hierarchical state-action structure caused by the agent's policy, the SI2E achieves enhanced maximum coverage exploration, thereby guaranteeing its exploration advantage.

**Estimation and Intrinsic Reward.** Considering the impracticality of directly acquiring visitation probabilities, we employ the $k$-NN entropy estimator in Equation 1 to estimate the lower bound:

$$H(V_0) - H(V_1) \approx \frac{d_z}{n_0} \cdot \sum_{i=0}^{n_0-1} \log d(v_i^0) - \frac{d_z}{n_1} \cdot \sum_{i=0}^{n_1-1} \log d(v_i^1) + C, \quad v_i^0 \in V_0, v_i^1 \in V_1, \tag{12}$$

where $d_z$ is the dimension of state-action embedding, $n_0$ and $n_1$ are the vertex numbers in $V_0$ and $V_1$, and $d(v)$ is twice the distance from vertex $v$ to its $k$-th nearest neighbor. By ignoring the constant term in Equation 12, we define the intrinsic reward $r_t^i$ and train RL agents to address the target task using a combined reward $r_t = r_t^e + \beta \cdot r_t^i$ (step II. f in Figure 2), where $\beta$ is a positive hyperparameter that modulates the trade-off between exploration and exploitation. The pseudocode, complexity analysis, and limitations of our framework are provided in Appendix A.

Table 1: Summary of success rates and required steps to achieve target rewards in MiniGrid and MetaWorld tasks: "average value ± standard deviation" and "average improvement". **Bold**: the best performance, underline: the second performance.

| MiniGrid | RedBlueDoors-6x6 | | SimpleCrossingS9N1 | | KeyCorridorS3R1 | |
|---|---|---|---|---|---|---|
| Navigation | Success Rate (%) | Required Step (K) | Success Rate (%) | Required Step (K) | Success Rate (%) | Required Step (K) |
| A2C | - | - | $88.18 \pm 3.46$ | $570.08 \pm 15.87$ | $86.57 \pm 2.26$ | $658.74 \pm 21.03$ |
| A2C+SE | - | - | $88.59 \pm 4.62$ | $394.39 \pm 66.14$ | $87.20 \pm 4.94$ | $463.86 \pm 38.27$ |
| A2C+VCSE | $79.82 \pm 7.26$ | $1161.90 \pm 241.59$ | $91.30 \pm 1.92$ | $204.02 \pm 25.60$ | $86.01 \pm 0.91$ | $190.20 \pm 6.11$ |
| A2C+SI2E | **$85.80 \pm 1.48$** | **$461.90 \pm 61.53$** | **$93.64 \pm 1.63$** | **$139.17 \pm 27.03$** | **$94.20 \pm 0.42$** | **$129.06 \pm 6.11$** |
| Abs.(%) Avg. | 5.98(7.49) ↑ | 700.0(60.25) ↓ | 2.34(2.56) ↑ | 64.85(31.79) ↓ | 7.00(8.03) ↑ | 61.14(32.15) ↓ |

| MiniGrid | DoorKey-6x6 | | DoorKey-8x8 | | Unlock | |
|---|---|---|---|---|---|---|
| Navigation | Success Rate (%) | Required Step (K) | Success Rate (%) | Required Step (K) | Success Rate (%) | Required Step (K) |
| A2C | $92.67 \pm 8.47$ | $567.20 \pm 96.57$ | - | - | $92.48 \pm 11.96$ | $669.78 \pm 154.74$ |
| A2C+SE | $93.18 \pm 6.81$ | $476.34 \pm 94.63$ | $72.60 \pm 20.32$ | $1515.81 \pm 324.28$ | $91.34 \pm 18.37$ | $634.37 \pm 240.51$ |
| A2C+VCSE | $94.08 \pm 2.58$ | $336.75 \pm 19.84$ | $94.32 \pm 11.09$ | $1900.96 \pm 398.65$ | $93.12 \pm 3.43$ | $405.22 \pm 52.22$ |
| A2C+SI2E | **$97.04 \pm 1.52$** | **$230.60 \pm 19.85$** | **$98.58 \pm 3.11$** | **$1090.96 \pm 125.77$** | **$97.13 \pm 3.35$** | **$309.14 \pm 53.71$** |
| Abs.(%) Avg. | 2.96(3.15) ↑ | 106.15(31.52) ↓ | 4.26(4.52) ↑ | 424.85(28.03) ↓ | 4.01(4.31) ↑ | 96.08(23.71) ↓ |

| MetaWorld | Button Press | | Door Open | | Drawer Open | |
|---|---|---|---|---|---|---|
| Manipulation | Success Rate (%) | Required Step (K) | Success Rate (%) | Required Step (K) | Success Rate (%) | Required Step (K) |
| DrQv2 | $94.55 \pm 4.64$ | $105.0 \pm 5.0$ | - | - | - | - |
| DrQv2+SE | $93.05 \pm 7.67$ | $95.0 \pm 5.0$ | - | - | $25.31 \pm 7.40$ | - |
| DrQv2+VCSE | $89.80 \pm 3.29$ | $77.5 \pm 2.5$ | $80.90 \pm 10.19$ | - | $82.74 \pm 7.46$ | $175.0 \pm 5.0$ |
| DrQv2+SI2E | **$99.60 \pm 0.57$** | **$62.5 \pm 7.5$** | **$95.77 \pm 1.05$** | **$87.5 \pm 2.5$** | **$95.96 \pm 3.00$** | **$82.5 \pm 2.5$** |
| Abs.(%) Avg. | 5.05(5.34) ↑ | 15.0(19.35) ↓ | 14.87(18.38) ↑ | - | 13.22(15.98) ↑ | 92.5(52.86) ↓ |

| MetaWorld | Faucet Close | | Faucet Open | | Window Open | |
|---|---|---|---|---|---|---|
| Manipulation | Success Rate (%) | Required Step (K) | Success Rate (%) | Required Step (K) | Success Rate (%) | Required Step (K) |
| DrQv2 | $53.33 \pm 1.92$ | - | - | - | $88.18 \pm 1.50$ | $192.5 \pm 2.5$ |
| DrQv2+SE | $92.36 \pm 3.66$ | $71.25 \pm 6.25$ | - | - | $93.14 \pm 2.03$ | $172.5 \pm 2.5$ |
| DrQv2+VCSE | $94.21 \pm 1.74$ | $60.0 \pm 5.0$ | $87.23 \pm 5.29$ | $67.5 \pm 5.0$ | $93.17 \pm 1.45$ | $127.5 \pm 7.5$ |
| DrQv2+SI2E | **$99.37 \pm 1.18$** | **$27.5 \pm 2.5$** | **$97.06 \pm 1.39$** | **$51.25 \pm 3.75$** | **$99.46 \pm 0.35$** | **$77.5 \pm 2.5$** |
| Abs.(%) Avg. | 5.16(5.48) ↑ | 32.5(54.17) ↓ | 9.83(11.27) ↑ | 16.25(24.07) ↓ | 6.29(6.75) ↑ | 50.0(39.22) ↓ |

## 5 Experiments

In this section, we present a comprehensive suite of comparative experiments on MiniGrid [Chevalier-Boisvert et al., 2018], MetaWorld [Yu et al., 2020], and the DeepMind Control Suite (DMControl) [Tunyasuvunakool et al., 2020] to evaluate the effectiveness of SI2E in terms of both final performance and sample efficiency. Consistent with previous work [Zeng et al., 2023c], we measure the required steps to attain specified rewards (0.9 times SI2E's convergence reward) as a benchmark for assessing sample efficiency. For the SI2E implementation, we employ a randomly initialized encoder optimized to minimize the combined loss $L$. All experiments are conducted with 10 different random seeds, and the learning curves are delineated in Appendix E.

### 5.1 MiniGrid Evaluation

Initially, we assess our framework on navigation tasks using the MiniGrid benchmark, which includes goal-reaching tasks in sparse-reward environments. This setting is partially observable: the agent receives a $7 \times 7 \times 3$ embedding of the immediate surrounding grid rather than the entire grid. For comparative purposes, we employ the A2C agent [Mnih et al., 2016] with Shannon entropy (SE) [Seo et al., 2021] and value-based state entropy (VCSE) [Kim et al., 2023] as our baselines. Table 1 (upper) displays the average values and standard deviations of success rates and required steps for various navigation tasks. The tasks encompass navigation with obstacles (SimpleCrossingS9N1), long-horizon navigation (RedBlueDoors, DoorKey, and Unlock), and long-horizon navigation with obstacles (KeyCorridorS3R1). The SI2E consistently exhibits enhanced final performance and sample efficiency across tasks, with an average success rate increase of $4.92\%$, from $89.97\%$ to $94.40\%$, and an average decrease in required steps of $38.10\%$, from $635.65K$ to $393.47K$. In the RedBlueDoors task, where baseline performances are inadequate, our SI2E significantly improves the success rate from $79.82\%$ to $85.80\%$ and reduces the required steps from $1161.90K$ to $461.90K$.

### 5.2 MetaWorld Evaluation

We further evaluate the SI2E framework on visual manipulation tasks from the MetaWorld benchmark, which presents exploration challenges due to its large state space. We select the model-free DrQv2 algorithm as the underlying RL methodology. Adhering to the setup of [Seo et al., 2023], we employ the same camera configuration and normalize the reward with a scale of 1. We summarize the success rates and required steps for all exploration methods across six MetaWorld tasks in Table 1 (lower).

Table 2: Summary of average episode rewards for control tasks in DMControl, encompassing two cartpole tasks characterized by sparse rewards: "average value $\pm$ standard deviation" and "average improvement" (absolute value(%)). **Bold**: the best performance, underline: the second performance.

| Domain, Task | Hopper Stand | Cheetah Run | Quadruped Walk | Pendulum Swingup | Cartpole Balance | Cartpole Swingup |
|---|---|---|---|---|---|---|
| DrQv2 | $87.59 \pm 11.70$ | $229.28 \pm 123.93$ | $289.79 \pm 24.17$ | $424.21 \pm 246.96$ | $998.97 \pm 22.95$ | $-$ |
| DrQv2+SE | $313.39 \pm 94.15$ | $228.82 \pm 126.21$ | $\underline{290.27} \pm 24.20$ | $10.80 \pm 2.92$ | $993.80 \pm 75.24$ | $219.69 \pm 62.21$ |
| DrQv2+VCSE | $711.32 \pm 30.84$ | $\underline{456.26} \pm 22.20$ | $243.74 \pm 29.91$ | $\underline{824.17} \pm 99.59$ | $\underline{998.65} \pm 9.58$ | $\underline{707.76} \pm 50.38$ |
| DrQv2+MADE | $\underline{717.09} \pm 112.94$ | $366.59 \pm 53.74$ | $262.63 \pm 23.92$ | $672.11 \pm 34.63$ | $996.16 \pm 40.60$ | $704.18 \pm 41.75$ |
| DrQv2+SI2E (Ours) | $\mathbf{797.17} \pm 53.21$ | $\mathbf{464.08} \pm 29.32$ | $\mathbf{399.51} \pm 29.05$ | $\mathbf{885.50} \pm 38.28$ | $\mathbf{999.58} \pm 2.97$ | $\mathbf{795.09} \pm 90.49$ |
| Abs.(%) Avg. ↑ | 80.08(11.17) | 7.82(1.71) | 109.24(37.63) | 61.33(7.44) | 0.93(0.09) | 87.33(12.34) |

Our SI2E framework enables the DrQv2 agent to solve all tasks with an average success rate of $97.87$ after an average of $64.79K$ environmental steps, significantly outperforming other baselines. Specifically, in the Door Open task, all baselines struggle to achieve a meaningful success rate with a satisfactory number of environmental steps. This result demonstrates the SI2E's effectiveness in improving and accelerating agent exploration in challenging tasks with expansive state-action spaces.

### 5.3 DMControl Evaluation

Subsequently, we evaluate our framework across various continuous control tasks within the DM-Control suite. As the foundational agent, we choose the same DrQv2 algorithm, which operates on pixel-based observations. We incorporate a state-action exploration baseline, MADE [Zhang et al., 2021b], for a more comprehensive comparison. We evaluate all exploration methods across six continuous control tasks, documenting the episode rewards in Table 2. Observations reveal that SI2E remarkably increases the mean episode reward in each DMControl task. Specifically, in the Cartpole Swingup task characterized by sparse rewards, our framework boosts the average reward from 707.76 to 795.09, resulting in a $12.34\%$ improvement in the final performance. Moreover, we compare the sample efficiency of SI2E and the best-performing baseline in Appendix E.3.

These results not only demonstrate the effectiveness of SI2E in acquiring dynamics-relevant representations for state-action pairs but also highlight its potential to motivate agents to explore the state-action space. To better understand the rationality and advantage of the SI2E framework, we provide visualization experiments in Appendix E.4.

### 5.4 Ablation Studies

To further investigate the impact of two critical components within the SI2E framework, embedding principle (Section 4.1) and intrinsic reward mechanism (4.2), we perform ablation studies on Meta-World and DMControl tasks, focusing on two distinct variants: (i) SI2E-DB, which utilizes the DB bottleneck [Bai et al., 2021] for learning state-action representations, and (ii) SI2E-VCSE, employing the state-of-the-art VCSE approach [Kim et al., 2023] for calculating intrinsic rewards. As depicted in Figure 3, SI2E surpasses all variants regarding final performance and sample efficiency. This outcome underscores the essential role of these critical components in conferring SI2E's superior capabilities. Additional ablation studies for the parameters $\beta$ and $n$ are available in Appendix E.5.

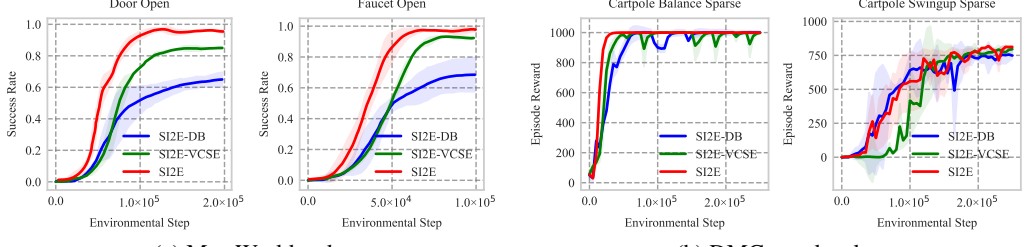

          (a) MetaWorld tasks                (b) DMControl tasks

Figure 3: Learning curves across MetaWorld and DMControl tasks for ablation studies.

## 6 Related Work

### 6.1 Maximum Entropy Exploration

Maximum entropy exploration has evolved from focusing initially on unsupervised methods to incorporating task rewards in more advanced supervised models. In the unsupervised paradigm,

agents autonomously acquire behaviors by using state entropy as an intrinsic reward for exploration [Liu and Abbeel, 2021, Mutti et al., 2022, Yang and Spaan, 2023]. In contrast, in the supervised paradigm, agents aim to maximize state entropy in conjunction with task rewards [Seo et al., 2021, Yuan et al., 2022].

However, these methods face challenges due to imbalances in the distributions of states with differing policy values. To address this issue, a value-based approach [Kim et al., 2023] has been proposed, which integrates value estimates into the entropy calculation to ensure balanced exploration. Nevertheless, the effectiveness of this approach heavily depends on the partitioning structure of states according to policy values, requiring prior knowledge about downstream tasks.

In this work, we leverage structural information principles to derive the hierarchical state-action structure in an unsupervised manner. We further define the value-conditional structural entropy as an intrinsic reward to achieve more effective agent exploration. Compared to current maximum entropy explorations, SI2E introduces an additional sub-community entropy and minimizes this entropy to motivate the agent to explore specific sub-communities with high policy values. This approach helps avoid redundant explorations within low-value sub-communities and achieves enhanced maximum coverage exploration. Our method allows for balanced exploration without requiring prior knowledge of downstream tasks, effectively addressing the limitations of previous approaches.

## 6.2 Representation Learning

Novelty Search [Tao et al., 2020] and Curiosity Bottleneck [Kim et al., 2019b] leverage the Information Bottleneck principle for effective representation learning. Additionally, the EMI method [Kim et al., 2019a] maximizes mutual information in both forward and inverse dynamics to develop desirable representations. However, these methods are limited by the lack of an explicit mechanism to address the white noise issue in the state space. To overcome this challenge, the Dynamic Bottleneck model [Bai et al., 2021] is introduced for robust exploration in complex environments.

Our work defines structural mutual information to measure the structural similarity between two variables for the first time. Additionally, we present an innovative embedding principle that incorporates the entropy of the representation variable. This approach more effectively eliminates irrelevant information than the traditional information bottleneck principle.

## 6.3 Structural Information Principles

Since the introduction of structural information principles [Li and Pan, 2016], these principles have significantly transformed the analysis of network complexities, employing metrics such as structural entropy and partitioning trees. This innovative approach has not only deepened the understanding of network dynamics—even in the context of multi-relational graphs [Cao et al., 2024a]—but has also led to a wide array of applications across different domains. The application of structural information principles has extended to various fields, including graph learning [Wu et al., 2022], skin segmentation [Zeng et al., 2023a], and the analysis of social networks [Peng et al., Zeng et al., 2024, Cao et al., 2024b]. In the domain of reinforcement learning, these principles have been instrumental in defining hierarchical action and state abstractions through encoding trees [Zeng et al., 2023b,c], marking a significant advancement in robust decision-making frameworks.

## 7 Conclusion

We propose SI2E, a novel exploration framework based on structural information principles. This framework defines structural mutual information to effectively capture state-action representations relevant to environmental dynamics. It maximizes the value-conditional structural entropy to enhance coverage across the state-action space. We have established theoretical connections between SI2E and traditional information-theoretic methodologies, underscoring the framework's rationality and advantages. Through extensive and comparative evaluations, SI2E significantly improves final performance and sample efficiency over state-of-the-art exploration methods. Our future work includes expanding the height of encoding trees and the range of experimental environments. Our goal is for SI2E to remain a robust and adaptable tool in reinforcement learning, particularly suited to high-dimensional and sparse-reward contexts.

## Acknowledgments

The corresponding authors are Hao Peng and Angsheng Li. This work is supported by the National Key R&D Program of China through grant 2021YFB1714800, NSFC through grants 61932002, 62322202, and 62432006, Beijing Natural Science Foundation through grant 4222030, Local Science and Technology Development Fund of Hebei Province Guided by the Central Government of China through grant 246Z0102G, Hebei Natural Science Foundation through grant F2024210008, and Guangdong Basic and Applied Basic Research Foundation through grant 2023B1515120020.

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

# A  Framework Details

## A.1  Notations

Table 3: Glossary of Notations.

| Notation | Description |
|---|---|
| $X; Y$ | Random variables/Vertex Sets |
| $x; y$ | Variable values/Vertices |
| $p$ | Probability |
| $H; I$ | Shannon entropy; Mutual information |
| $\mathcal{O}; \mathcal{A}$ | Observation space; Action space |
| $O; S; A$ | Observation, state, action variables/Vertex sets |
| $o; s; a$ | Single observation, state, action/vertex |
| $z; Z$ | Single embedding; Embedding variables |
| $\mathcal{P}$ | Transition function |
| $r; \mathcal{R}$ | Reward; Reward function |
| $\pi; \gamma; t$ | Policy network; Discount factor; Timestep |
| $f; q$ | Encoder; Decoder |
| $G$ | Graph |
| $v; V$ | Single vertex; Vertex set |
| $e; E; w$ | Edge; Edge set; Edge weight |
| $\alpha; T$ | Tree node; Encoding tree |
| $\lambda; \nu$ | Root node; Leaf node |
| $H^T, H^K, H^{SI}; \Delta H$ | Structural entropy; Entropy reduction |
| $\mathcal{T}$ | Approximate binary trees |
| $I^{SI}$ | Structural mutual information |

## A.2  Tree Optimization on $\mathcal{T}^2$

In this subsection, we have provided additional explanations and illustrative examples for the encoding tree optimization on $\mathcal{T}^2$. As shown in Figure 4, the stretch operator is executed over sibling nodes $\alpha_i$ and $\alpha_j$ that share the same parent node, $\lambda$. The detailed steps of this operation are as follows:

$$\alpha'^- = \lambda, \quad \alpha_i^- = \alpha', \quad \alpha_j^- = \alpha', \tag{13}$$

where $\alpha'$ is the added tree node via the stretch operation.

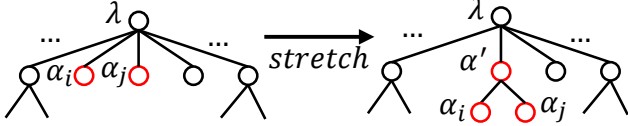

Figure 4: Stretch operation on sibling nodes within the encoding tree.

The corresponding variation in structural entropy, $\Delta H$, due to the stretch operation is calculated as:

$$H^T(G; \alpha_i) = -\frac{g_{\alpha_i}}{\text{vol}(G)} \cdot \log \frac{\text{vol}(\alpha_i)}{\text{vol}(G)}, \quad H^T(G; \alpha_j) = -\frac{g_{\alpha_j}}{\text{vol}(G)} \cdot \log \frac{\text{vol}(\alpha_j)}{\text{vol}(G)}, \tag{14}$$

$$H^{T'}(G; \alpha_i) = -\frac{g_{\alpha_i}}{\text{vol}(G)} \cdot \log \frac{\text{vol}(\alpha_i)}{\text{vol}(\alpha')}, \quad H^{T'}(G; \alpha_j) = -\frac{g_{\alpha_j}}{\text{vol}(G)} \cdot \log \frac{\text{vol}(\alpha_j)}{\text{vol}(\alpha')}, \tag{15}$$

$$H^{T'}(G; \alpha') = -\frac{g_{\alpha'}}{\text{vol}(G)} \cdot \log \frac{\text{vol}(\alpha')}{\text{vol}(G)}, \tag{16}$$

$$\begin{aligned}
\Delta H &= H^T(G; \alpha_i) + H^T(G; \alpha_j) - H^{T'}(G; \alpha_i) - H^{T'}(G; \alpha_j) - H^{T'}(G; \alpha') \\
&= (H^T(G; \alpha_i) - H^{T'}(G; \alpha_i)) + (H^T(G; \alpha_j) - H^{T'}(G; \alpha_j)) - H^{T'}(G; \alpha') \\
&= -\frac{g_{\alpha_i}}{\text{vol}(G)} \cdot \log \frac{\text{vol}(\alpha')}{\text{vol}(G)} - \frac{g_{\alpha_j}}{\text{vol}(G)} \cdot \log \frac{\text{vol}(\alpha')}{\text{vol}(G)} + \frac{g_{\alpha'}}{\text{vol}(G)} \cdot \log \frac{\text{vol}(\alpha')}{\text{vol}(G)} \\
&= -\frac{g_{\alpha_i} + g_{\alpha_j} - g_{\alpha'}}{\text{vol}(G)} \cdot \log \frac{\text{vol}(\alpha')}{\text{vol}(G)}.
\end{aligned} \tag{17}$$

As shown in Algorithm 1, the HCSE algorithm iteratively and greedily selects the pair of sibling nodes that cause the maximum entropy variation, $\Delta H$ to execute one stretch optimization.

---

**Algorithm 1** The Encoding Tree Optimization on $\mathcal{T}^2$

---

1: **Input:** one-layer initial encoding tree $T$
2: **Output:** the optimal encoding tree $T^* \in \mathcal{T}^2$
3: **while** True **do**
4:     $(\alpha_i, \alpha_j) \leftarrow$ maximize the entropy reduction $\Delta H$ caused by one stretch operation
5:     **if** $\Delta H = 0$ **then**
6:         Break
7:     **end if**
8:     Create a new tree node $\alpha'$
9:     $(\alpha')^- \leftarrow \lambda, (\alpha_i)^- \leftarrow \alpha', (\alpha_j)^- \leftarrow \alpha'$
10: **end while**

---

### A.3 Intuitive Example of Optimal Encoding Tree

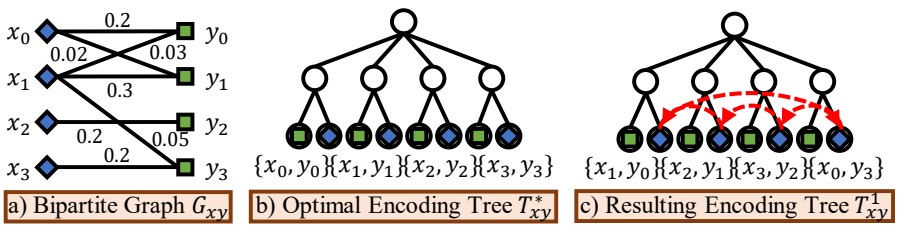

a) Bipartite Graph $G_{xy}$    b) Optimal Encoding Tree $T^*_{xy}$    c) Resulting Encoding Tree $T^1_{xy}$

Figure 5: Illustration from a joint distribution to 2-layer approximate binary trees: a) Bipartite distribution graph, b) Optimal encoding tree, c) Resulting encoding tree via a 1-transformation.

## A.4 The Pseudocode of SI2E

---

**Algorithm 2** Effective Exploration based on Structural Information Principles

---

1: **Input:** batch size $n$, update interval $t_{\text{up}}$
2: **Initialize:** agent's policy $\pi$, encoder functions $f_s$ and $f_z$, decoder functions $q_m$, $q_{z|s}$, $q_{s|z}$, replay buffer $\mathcal{B}$
3: **for** each episode **do**
4:     **for** each environmental step $t$ **do**
5:         Collect transition $\tau_t = (s_t, a_t, s_{t+1}, r_t^{\text{e}})$ using the encoder $f_s$ and policy $\pi$
6:         Sample a batch $\{\tau_{t_i}\}_{i=1}^n$ from $\mathcal{B}$ including the variables $S_t$, $A_t$, and $S_{t+1}$
7:         Adopt encoder functions $f_z$ to obtain state-action embeddings $Z_t$
8:         ***# Maximum Structural Entropy Exploration***
9:         Construct the state-action graph $G_{sa}$ according to the policy $\pi$ and generate its hierarchical community structure $T_{sa}^*$
10:         Employ the k-NN estimator to estimate the lower bound $H(V_0) - H(V_1)$ and compute intrinsic reward $r_t^i$
11:         Compute total reward $r_t = r_t^{\text{e}} + \beta \cdot r_t^{\text{i}}$
12:         Update $\tau_t' = (s_t, a_t, s_{t+1}, r_t)$ and augment $\mathcal{B}$ with $\tau_t'$
13:         **if** $t \mod t_{\text{up}} = 0$ **then**
14:             ***# Structural Mutual Information Principle***
15:             Compute representation losses $L_{\text{up}}$, $L_{z|s}$, and $L_{s|z}$
16:             Update encoder and decoder functions to minimize the combined loss $L$
17:             Update agent policy $\pi$ using $\mathcal{B}$
18:         **end if**
19:     **end for**
20: **end for**

---

## A.5 Complexity Analysis of SI2E

Within the SI2E framework, we analyze the time complexities of critical components independent of the underlying RL algorithm. During the state-action representation phase, the construction of bipartite graphs takes $O(n^2)$ time complexity, the generation of 2-layer approximate binary trees requires $O(n \cdot \log^2 n)$ time complexity, and the calculation of mutual information involves a time complexity of $O(n^2)$. During the effective exploration phase, the generation of hierarchical community structure incurs a $O(n \cdot \log^2 n)$ complexity, the construction of the distribution graph leads to a complexity of $O(n^2)$, and value-conditional structural entropy is calculated with $O(n)$ time complexity.

## A.6 Limitations

Our work, which is a result of thorough research, aims to address the limitations of information theory methods and structural information theory research in reinforcement learning. Therefore, we have selected the state-of-the-art information theory exploration method as the baseline in our evaluation. Despite the current limitations in height due to complexity and cost issues, the encoding tree structure in the SI2E framework holds immense potential. In our future research, we are optimistic about expanding its height further and conducting more research on the advantages and restrictions brought by this expansion.

# B  Theorem Proofs

## B.1  Proof of Proposition 3.1

*Proof.* For any two vertices $v_i \in V$ and $v_j \in V$ without any edge connecting them, their corresponding tree nodes are denoted as $\alpha_i$ and $\alpha_j$. These nodes' parents are initially assigned as the root node $\lambda$. Before executing one stretch operation on vertices $\alpha_i$ and $\alpha_j$, their structural entropies are calculated as follows:

$$H(G; \alpha_i) = -\frac{d_i}{\text{vol}(G)} \cdot \log \frac{d_i}{\text{vol}(G)}, \quad H(G; \alpha_j) = -\frac{d_j}{\text{vol}(G)} \cdot \log \frac{d_j}{\text{vol}(G)}, \tag{18}$$

where $d_i$ and $d_j$ are the degrees of vertices $v_i$ and $v_j$. Post-stretch operation, their structural entropies are given by:

$$H(G; \alpha_i) = -\frac{d_i}{\text{vol}(G)} \cdot \log \frac{d_i}{\text{vol}(\alpha')}, \quad H(G; \alpha_j) = -\frac{d_j}{\text{vol}(G)} \cdot \log \frac{d_j}{\text{vol}(\alpha')}, \tag{19}$$

where $\alpha'$ are their new common parent node. The absence of an edge between $v_i$ and $v_j$ ensures that:

$$g_{\alpha'} = d_i + d_j, \quad \text{vol}(\alpha') = d_i + d_j. \tag{20}$$

The structural entropy of $\alpha'$ can be determined as:

$$H(G; \alpha') = -\frac{g_{\alpha'}}{\text{vol}(G)} \cdot \log \frac{\text{vol}(\alpha')}{\text{vol}(G)} = -\frac{d_i + d_j}{\text{vol}(G)} \cdot \log \frac{d_i + d_j}{\text{vol}(G)}. \tag{21}$$

The entropy reduction $\Delta H$, consequent to the stretch operation on vertices $v_i$ and $v_j$, is calculated as:

$$\begin{aligned}
\Delta H &= \left[ -\frac{d_i}{\text{vol}(G)} \cdot \log \frac{d_i}{\text{vol}(G)} - \frac{d_j}{\text{vol}(G)} \cdot \log \frac{d_j}{\text{vol}(G)} \right] \\
&\quad - \left[ -\frac{d_i}{\text{vol}(G)} \cdot \log \frac{d_i}{d_i + d_j} - \frac{d_j}{\text{vol}(G)} \cdot \log \frac{d_j}{d_i + d_j} - \frac{d_i + d_j}{\text{vol}(G)} \cdot \log \frac{d_i + d_j}{\text{vol}(G)} \right] \\
&= \left[ -\frac{d_i}{\text{vol}(G)} \cdot \log \frac{d_i}{\text{vol}(G)} - \frac{d_j}{\text{vol}(G)} \cdot \log \frac{d_j}{\text{vol}(G)} \right] - \left[ -\frac{d_i}{\text{vol}(G)} \cdot \log \frac{d_i}{\text{vol}(G)} - \frac{d_j}{\text{vol}(G)} \cdot \log \frac{d_j}{\text{vol}(G)} \right] \\
&= 0.
\end{aligned} \tag{22}$$

Given the zero reduction in entropy, as per lines 5 and 6 of the optimization algorithm for $\mathcal{T}^2$ (See Appendix A.2), the stretch operation involving $v_i$ and $v_j$ is omitted from the optimization process. $\square$

## B.2  Proof of Theorem 3.4

*Proof.* The difference between the mutual information $I^{SI}(X; Y)$ and $I(X; Y)$ is expressed as:

$$\begin{aligned}
I^{SI}(X; Y) - I(X; Y) &= \sum_{i,j} \left[ p(x_i, y_j) \cdot \log \frac{2}{p(x_i) + p(y_j)} \right] - \sum_{i,j} \left[ p(x_i, y_j) \cdot \log \frac{p(x_i, y_j)}{p(x_i) \cdot p(y_j)} \right] \\
&= \sum_{i,j} \left[ p(x_i, y_j) \cdot \log \left[ \frac{2}{p(x_i) + p(y_j)} \cdot \frac{p(x_i) \cdot p(y_j)}{p(x_i, y_j)} \right] \right] \\
&= \sum_{i,j} \left[ p(x_i, y_j) \cdot \log \left[ \frac{1}{p(x_i, y_j)} \cdot \frac{2 \cdot p(x_i) \cdot p(y_j)}{p(x_i) + p(y_j)} \right] \right] \\
&= \sum_{i,j} \left[ p(x_i, y_j) \cdot \log \left[ \frac{1}{p(x_i, y_j)} \cdot \frac{2}{\frac{1}{p(x_i)} + \frac{1}{p(y_j)}} \right] \right].
\end{aligned} \tag{23}$$

Given the conditions $p(x_i, y_j) \leq p(x_i) \leq 1$ and $p(x_i, y_j) \leq p(y_j) \leq 1$, the following inequalities are satisfied:

$$\frac{1}{p(x_i)} + \frac{1}{p(y_j)} \leq \frac{2}{p(x_i, y_j)}, \quad \frac{1}{p(x_i, y_j)} \cdot \frac{2}{\frac{1}{p(x_i)} + \frac{1}{p(y_j)}} \geq 1, \tag{24}$$

$$0 \leq \log_{p(x_i,y_j)} p(x_i) \leq 1, \quad 0 \leq \log_{p(x_i,y_j)} p(y_j) \leq 1. \tag{25}$$

By defining $\epsilon_i = \log_{p(x_i,y_j)} p(x_i)$ and $\epsilon_j = \log_{p(x_i,y_j)} p(y_j)$, we obtain the following results:

$$
\begin{aligned}
I^{SI}(X;Y) - I(X;Y) &= \sum_{i,j} \left[ p(x_i,y_j) \cdot \log \left[ \frac{1}{p(x_i,y_j)} \cdot \frac{2}{\left[\frac{1}{p(x_i,y_j)}\right]^{\epsilon_i} + \left[\frac{1}{p(x_i,y_j)}\right]^{\epsilon_j}} \right] \right] \\
&\leq (1 - \min(\epsilon_i, \epsilon_j)) \cdot \sum_{i,j} \left[ p(x_i,y_j) \cdot \log \frac{1}{p(x_i,y_j)} \right] \\
&= (1 - \min(\epsilon_i, \epsilon_j)) \cdot H(X,Y).
\end{aligned}
\tag{26}
$$

Therefore,

$$0 \leq I^{SI}(X;Y) - I(X;Y) \leq (1 - \min(\epsilon_i, \epsilon_j)) \cdot H(X,Y), \tag{27}$$

$$I(X;Y) \leq I^{SI}(X;Y) \leq I(X;Y) + (1 - \min(\epsilon_i, \epsilon_j)) \cdot H(X,Y). \tag{28}$$

$\square$

## B.3 Proof of Theorem 4.1

*Proof.* For a one-to-one correspondence of variables $X$ and $Y$, the joint probability of a tuple $(x_i, y_j)$ is as follows:

$$
p(x_i, y_j) = \begin{cases} p(x_i) = p(y_j) & \text{if } i = j, \\ 0 & \text{otherwise.} \end{cases}
\tag{29}
$$

The calculation for $I^{SI}(X;Y)$ is carried out in the following manner:

$$
\begin{aligned}
I^{SI}(X;Y) &= \sum_{l=0}^{n} \left[ H^{SI}(X) + H^{SI}(Y) - H^{T_{xy}^l}(X,Y) \right] \\
&= \sum_{i,j} \left[ p(x_i, y_j) \cdot \log \frac{2}{p(x_i) + p(y_j)} \right] \\
&= \sum_{i} \left[ p(x_i) \cdot \log \frac{1}{p(x_i)} \right] \\
&= H(X).
\end{aligned}
\tag{30}
$$

Similarly, the calculation for $I(X;Y)$ proceeds as follows:

$$
\begin{aligned}
I(X;Y) &= \sum_{i,j} \left[ p(x_i, y_j) \cdot \log \frac{p(x_i, y_j)}{p(x_i) \cdot p(y_j)} \right] \\
&= \sum_{i} \left[ p(x_i) \cdot \log \frac{1}{p(x_i)} \right] \\
&= H(X).
\end{aligned}
\tag{31}
$$

Hence, $I^{SI}(X;Y)$ equals $I(X;Y)$ when the joint distribution between $X$ and $Y$ shows a one-to-one correspondence. $\square$

## B.4 Proof of Proposition 4.2

*Proof.* Now, we employ mathematical induction to demonstrate the existence of the graph $G'_{sa}$.
**Base Case ($n = 2$):** Suppose the degree distribution of two vertices is given by $(p_0, p_1)$ with $p_0 \leq p_1$.
We construct the graph $G'_{sa}$ as follows:
• Create an edge with weight $p_0$ between $v_0$ and $v_1$.
• Add a self-connected edge at vertex $v_1$ with weight $p_1 - p_0$.
**Inductive Step ($n = k$):** Assume that, the graph $G'_{sa}$ with $k$ vertices exists and satisfies Proposition 4.2.
**Inductive Case ($n = k + 1$):** Consider the addition of a new vertex $v_k$ to construct $G'_{sa}$ with $k + 1$

vertices using the following steps:

• Start with a subgraph that includes the first $k$ vertices, yielding a distribution of $(p'_0, \ldots, p'_{k-1})$ with $\sum_{i=0}^{k-1} p'_i = 1$.

• Modify the weight of all edges connected to each vertex $v_i$ ($0 \leq i \leq k$) by a factor $\frac{(1-p_n)p_i}{p'_i}$.

• For each vertex $v_i$, create an edge with weight $p_i p_n$ connecting it to the new vertex $v_k$.

• Add a self-connected edge at vertex $v_k$ with weight $p_k^2$.

These modifications ensure that the degree distribution of the graph remains consistent with the addition of the new vertex, thus completing the inductive step and proving the existence of $G'_{sa}$ for any $n$. □

## B.5 Proof of Theorem 4.3

In the tree $T^*_{sa}$, we denote the $i$-th intermediate node as $\alpha_i$ and its $j$-th child node as $\alpha_{ij}$. The single state-action vertex in the corresponding subset of $\alpha_{ij}$ is assumed as $(s_{ij}, a_{ij})$. In the graph $G'_{sa}$, the degree of any state-action vertex $(s_{ij}, a_{ij})$ is equated to its visitation probability, thereby:

$$\text{vol}(G'_{sa}) = \sum_{i,j} p(s_{ij}, a_{ij}) = 1, \quad \text{vol}(\alpha_{ij}) = g_{\alpha_{ij}} = p(s_{ij}, a_{ij}). \tag{32}$$

The 2-dimensional value-conditional structural entropy $H^{T^*_{sa}}(G'_{sa})$ is calculated through the following expressions:

$$
\begin{aligned}
H^{T^*_{sa}}(G'_{sa}) &= -\sum_i \left[ g_{\alpha_i} \cdot \log \text{vol}(\alpha_i) + \sum_j \left[ g_{\alpha_{ij}} \cdot \log \frac{\text{vol}(\alpha_{ij})}{\text{vol}(\alpha_i)} \right] \right] \\
&= -\sum_i \left[ g_{\alpha_i} \cdot \log \text{vol}(\alpha_i) + \sum_j \left[ p(s_{ij}, a_{ij}) \cdot \log \frac{p(s_{ij}, a_{ij})}{\text{vol}(\alpha_i)} \right] \right] \\
&= \sum_{i,j} \left[ p(s_{ij}, a_{ij}) \cdot \log \frac{1}{p(s_{ij}, a_{ij})} \right] - \sum_i \left[ \text{vol}(\alpha_i) \cdot \log \frac{1}{\text{vol}(\alpha_i)} \right] + \sum_i \left[ g_{\alpha_i} \cdot \log \frac{1}{\text{vol}(\alpha_i)} \right] \\
&= H(V_0) - H(V_1) + \sum_i \left[ g_{\alpha_i} \cdot \log \frac{1}{\text{vol}(\alpha_i)} \right] \\
&\geq H(V_0) - H(V_1).
\end{aligned}
\tag{33}
$$

Given that $p(s_{ij}, a_{ij}) \cdot \text{vol}(\alpha_i) \leq p(s_{ij}, a_{ij}) \leq \text{vol}(\alpha_i)$, this inequalities hold that:

$$0 \leq \log_{p(s_{ij}, a_{ij})} \frac{p(s_{ij}, a_{ij})}{\text{vol}(\alpha_i)} \leq 1. \tag{34}$$

By defining $\zeta_{ij} = \log_{p(s_{ij}, a_{ij})} \frac{p(s_{ij}, a_{ij})}{\text{vol}(\alpha_i)}$, we obtain that:

$$
\begin{aligned}
\text{vol}(\alpha_i) \cdot \log \frac{1}{\text{vol}(\alpha_i)} &= \sum_j p(s_{ij}, a_{ij}) \cdot \frac{1}{\text{vol}(\alpha_i)} \\
&= \sum_j p(s_{ij}, a_{ij}) \cdot \log \frac{1}{[p(s_{ij}, a_{ij})]^{1-\zeta_{ij}}} \\
&= \sum_j \left[ (1 - \zeta_{ij}) \cdot p(s_{ij}, a_{ij}) \cdot \log \frac{1}{p(s_{ij}, a_{ij})} \right].
\end{aligned}
\tag{35}
$$

Selecting the minimal $\zeta$-value as $\zeta^*$ allow us to reformulate Equation 33 as follows:

$$H(V_1) = \sum_i \left[ (\text{vol}(\alpha_i)) \cdot \log \frac{1}{\text{vol}(\alpha_i)} \right] \leq (1 - \zeta^*) \cdot H(S_t, A_t), \tag{36}$$

$$H(V_0) = H(S_t, A_t), \tag{37}$$

$$\zeta^* \cdot H(S_t, A_t) \leq H(V_0) - H(V_1) \leq H^{T^*_{sa}}(G'_{sa}) \leq H(S_t, A_t). \tag{38}$$

## C Detailed Derivations

### C.1 Derivation of $I^{SI}(X;Y)$

For each intermediate node $\alpha_i \in T^*_{xy}$ with vertex subset $\{x_i, y_i\}$, the entropy sum of this node and its children is calculated as follows:

$$-\frac{g_{\alpha_i}}{\text{vol}(G_{xy})} \cdot \log \frac{\text{vol}(\alpha_i)}{\text{vol}(G_{xy})} - \frac{p(x_i)}{\text{vol}(G_{xy})} \cdot \log \frac{p(x_i)}{\text{vol}(\alpha_i)} - \frac{p(y_i)}{\text{vol}(G_{xy})} \cdot \log \frac{p(y_i)}{\text{vol}(\alpha_i)}$$

$$= -\frac{g_{\alpha_i}}{2} \cdot \log \frac{\text{vol}(\alpha_i)}{2} - \frac{p(x_i)}{2} \cdot \log \frac{p(x_i)}{\text{vol}(\alpha_i)} - \frac{p(y_i)}{2} \cdot \log \frac{p(y_i)}{\text{vol}(\alpha_i)}. \tag{39}$$

The $H^{SI}(X) + H^{SI}(Y) - H^{T^*_{xy}}(X,Y)$ term in Equation 5 is calculated as follows:

$$\sum_i \left[ -\frac{p(x_i)}{2} \cdot \log \frac{p(x_i)}{2} - \frac{p(y_i)}{2} \cdot \log \frac{p(y_i)}{2} \right] - \sum_i \left[ -\frac{g_{\alpha_i}}{2} \cdot \log \frac{\text{vol}(\alpha_i)}{2} - \frac{p(x_i)}{2} \cdot \log \frac{p(x_i)}{\text{vol}(\alpha_i)} - \frac{p(y_i)}{2} \cdot \log \frac{p(y_i)}{\text{vol}(\alpha_i)} \right]. \tag{40}$$

Proposition 3.1 ensures that:

$$\text{vol}(\alpha_i) = p(x_i) + p(y_i), \quad g_{\alpha_i} = p(x_i) + p(y_i) - 2p(x_i, y_i). \tag{41}$$

Consequently, we can reformulate Equation 40 as follows:

$$\sum_i \left[ \frac{p(x_i)}{2} \cdot \log \frac{2}{p(x_i) + p(y_i)} + \frac{p(y_i)}{2} \cdot \log \frac{2}{p(x_i) + p(y_i)} - \frac{p(x_i) + p(y_i) - 2 \cdot p(x_i, y_i)}{2} \cdot \log \frac{2}{p(x_i) + p(y_i)} \right]$$

$$= \sum_i \left[ p(x_i, y_i) \cdot \log \frac{2}{p(x_i) + p(y_i)} \right]. \tag{42}$$

For any integer $l > 0$, the $H^{SI}(X) + H^{SI}(Y) - H^{T^l_{xy}}(X,Y)$ term in Equation 5 is given by:

$$\sum_i \left[ p(x_{i'}, y_i) \cdot \log \frac{2}{p(x_{i'}) + p(y_i)} \right], \quad i' = (i + l) \bmod n. \tag{43}$$

Consequently,

$$I^{SI}(X;Y) = \sum_{l=0}^{n} \left[ H^{SI}(X) + H^{SI}(Y) - H^{T^l_{xy}}(X,Y) \right] = \sum_{i,j} \left[ p(x_i, y_j) \cdot \log \frac{2}{p(x_i) + p(y_j)} \right]. \tag{44}$$

### C.2 Upper Bound of $I^{SI}(Z_t; S_t)$

Theorem 3.4 assures the following inequality:

$$I^{SI}(X;Y) \leq I(X;Y) + (1 - \epsilon) \cdot H(X,Y), \quad 0 \leq \epsilon \leq 1. \tag{45}$$

Given the relationship between joint entropy and conditional entropy in traditional information theory, this inequality can be reformulated as follows:

$$\begin{aligned} I^{SI}(X;Y) &\leq I(X;Y) + (1 - \epsilon) \cdot H(X,Y) \\ &= I(X;Y) + (1 - \epsilon) \cdot H(X|Y) + (1 - \epsilon) \cdot H(Y) \\ &\leq I(X;Y) + H(X|Y) + H(Y). \end{aligned} \tag{46}$$

### C.3 Upper Bound of $I(Z_t; S_t)$

Through the non-negativity of KL-divergence, the following upper bound of $I(Z_t; S_t)$ holds that:

$$\begin{aligned} I(Z_t; S_t) &= \sum \left[ p(z_t, s_t) \cdot \log \frac{p(z_t|s_t)}{p(z_t)} \right] \\ &= \sum \left[ p(z_t, s_t) \cdot \log \frac{p(z_t|s_t)}{q_m(z_t)} \right] - D_{KL}(p||q_m) \\ &\leq \sum [p(z_t, s_t) \cdot D_{KL}(p(z_t|s_t)||q_m(z_t))]. \end{aligned} \tag{47}$$

## C.4 Upper Bound of $H(Z_t|S_t)$

Through the non-negativity of KL-divergence, the following upper bound of $H(Z_t|S_t)$ holds that:

$$
\begin{aligned}
H(Z_t|S_t) &= \sum \left[ p(z_t, s_t) \cdot \log \frac{1}{p(z_t|s_t)} \right] \\
&= \sum \left[ p(z_t, s_t) \cdot \log \frac{1}{q_{z|s}(z_t|s_t)} \right] - D_{KL}(p||q_{z|s}) \\
&\leq \sum \left[ p(z_t, s_t) \cdot \log \frac{1}{q_{z|s}(z_t|s_t)} \right].
\end{aligned}
\tag{48}
$$

## C.5 Lower Bound of $I(Z_t; S_{t+1})$

Leveraging the non-negative Shannon entropy and KL-divergence, we obtain the lower bound of $I(Z_t; S_{t+1})$:

$$
\begin{aligned}
I(Z_t; S_{t+1}) &= \sum \left[ p(z_t, s_{t+1}) \cdot \log \frac{p(s_{t+1}|z_t)}{p(s_{t+1})} \right] \\
&= \sum \left[ p(z_t, s_{t+1}) \cdot \log q_{s|z}(s_{t+1}|z_t) \right] + H(S_{t+1}) + D_{KL}(p||q_{s|z}) \\
&\geq \sum \left[ p(z_t, s_{t+1}) \cdot \log q_{s|z}(s_{t+1}|z_t) \right].
\end{aligned}
\tag{49}
$$

# D  Experimental Details

In the experiments conducted for this work, we utilize a single NVIDIA RTX A1000 GPU and eight Intel Core i9 CPU cores clocked at 3.00GHz for each training run. The total number of environmental steps was set to $3000K/1000K$ for the MiniGrid benchmark, $200K/100K$ for the MetaWorld benchmark, and $250K$ for the DeepMind Control Suite (DMControl).

## D.1  Implementation Details

**A2C implementation details.** In our implementation of the A2C algorithm, we utilize the official RE3 implementation[2], adhering to the pre-established hyperparameters set, except where explicitly noted. State representations in baselines are derived using a fixed encoder that is randomly initialized, and intrinsic rewards are normalized based on the standard deviation computed from sample data, consistent with the original methodology. However, this normalization process is omitted in the intrinsic reward calculations for VCSE and SI2E implementations. Across all exploration methods, we maintain fixed scale parameters $\beta = 0.005$ and $k = 5$, in line with the original framework. The comprehensive hyperparameters for the A2C algorithm are detailed in Table 4.

Table 4: Hyperparameters for the A2C algorithm on the MiniGrid benchmark.

| Hyperparameter | Value |
|---|---|
| number of updates between two savings | 100 |
| number of processes | 16 |
| number of frames in training | $3e6/1e6$ |
| scale parameter $\beta$ | 0.005 |
| batch size | 256 |
| number of frames per process before update | 5 |
| discount factor | 0.99 |
| learning rate | 0.001 |
| GAE coefficient | 0.95 |
| maximum norm of gradient | 0.5 |

**DrQv2 implementation details.** For the DrQv2 algorithm, we employ its official implementation[3] [Yarats et al., 2021], maintaining the original hyperparameter settings unless specified otherwise. A fixed noise level of $0.2$ and $k = 12$ are used for all exploration methods, including SE, VCSE, MADE, and SI2E. In the calculation of intrinsic rewards in baselines, we train the Intrinsic Curiosity Module [Pathak et al., 2017] using representations from the visual encoder to measure vertex distance in the estimation of value-conditional structural entropy. Specific hyperparameters in the DrQv2 are summarized in Table 5.

Table 5: Hyperparameters for the DrQv2 algorithm on the DeepMind Control Suite.

| Hyperparameter | Value |
|---|---|
| number of frames stacked | 3 |
| number of times each action is repeated | 2 |
| number of frames for an evaluation | 10000 |
| number of episodes for each evaluation | 10 |
| number of worker threads for the replay buffer | 4 |
| replay buffer size | $1e6$ |
| batch size | 64 |
| discount factor | 0.99 |
| learning rate | 0.0001 |
| feature dimensionality | 50 |
| hidden dimensionality | 1024 |
| scale parameter $\beta$ | 0.1 |

---

[2] https://github.com/younggyoseo/RE3
[3] https://github.com/facebookresearch/drqv2

## D.2 Environment Details

**MiniGrid Experiments.** In our MiniGrid benchmark experiments, we encompass six navigation tasks, including RedBlueDoors, SimpleCorssingS9N1, KeyCorridor, DoorKey-6x6, DoorKey-8x8, and Unlock, with visual representations provided in Figure 6. Notably, all tasks are employed in their original forms without any modifications.

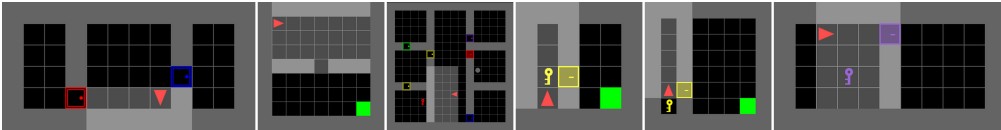

Figure 6: Examples of navigation tasks used in our MiniGrid experiments include: (a) RedBlueDoors, (b) SimpleCorssingS9N1, (c) KeyCorridor, (d) DoorKey-6x6, (e) DoorKey-8x8, (f) Unlock.

**MetaWorld Experiments.** In our evaluation using the MetaWorld benchmark, we conduct experiments on six manipulation tasks: Door Open, Drawer Open, Faucet Open, Window Open, Button Press, and Faucet Close. These tasks are visualized in Figure 7.

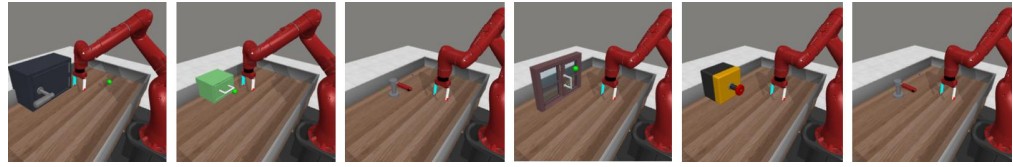

Figure 7: Examples of manipulation tasks used in our MetaWorld experiments include: (a) Door Open, (b) Drawer Open, (c) Faucet Open, (d) Window Open, (e) Button Press, (f) Faucet Close.

**DMControl Experiments.** Our research in DMControl suite focuses on six continuous control tasks, specifically Hopper Stand, Cheetah Run, Quadruped Walk, Pendulum Swingup, Cartpole Balance Sparse, and Cartpole Swingup Sparse. And visualizations of thes tasks are provided in Figure 8.

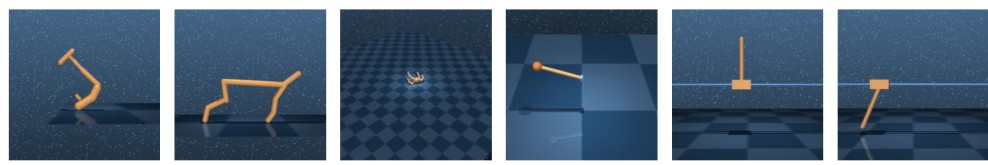

Figure 8: Examples of control tasks used in our DeepMind Control Suite experiments include: (a) Hopper Stand, (b) Cheetah Run, (c) Quadruped Walk, (d) Pendulum Swingup, (e) Cartpole Balance Sparse, (f) Cartpole Swingup Sparse.

# E Additional Experiments

## E.1 Experiments on MiniGrid Benchmark

Figure 9 illustrates the learning curves for the A2C algorithm integrated with our SI2E framework, as well as with other exploration baselines, SE and VCSE. The corresponding variants are labeled as A2C, A2C+SE, A2C+VCSE, and A2C+SI2E. These results demonstrate that SI2E consistently outperforms other baselines across various navigation tasks. In terms of final performance, A2C+SI2E achieves an average success rate of $94.40\%$ across all navigation tasks, significantly outperforming the second-best A2C+VCSE, which has an average success rate of $89.78\%$. Regarding sample efficiency, SI2E converges in fewer than $50\%$ of the total environmental steps in almost all tasks, except for DoorKey-8x8. This indicates SI2E's effectiveness in enhancing the agent's exploration of the state-action space, surpassing the baseline methods.

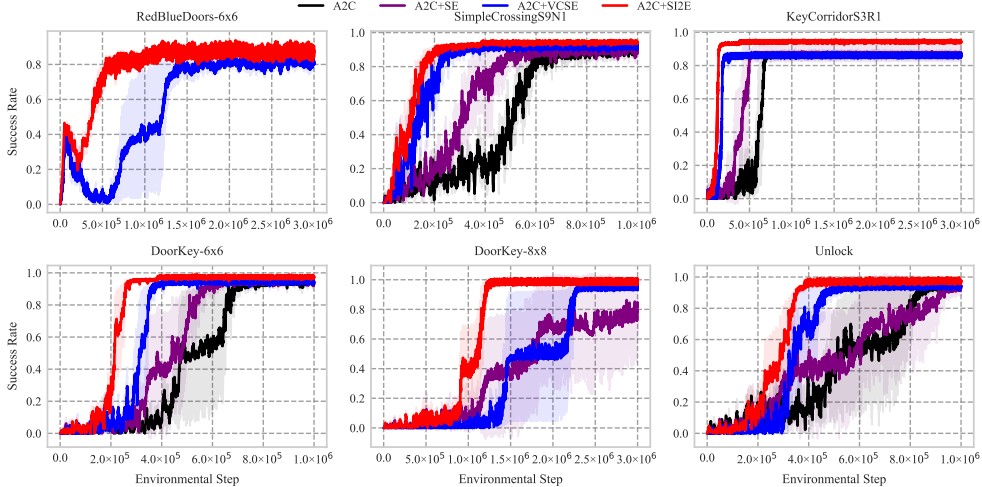

Figure 9: Learning curves for six navigation tasks in MiniGrid, measured in terms of success rate. The solid lines represent the interquartile mean, while the shaded regions indicate the standard deviation, both calculated across 10 runs.

To further substantiate our results on the Minigrid environment, we have introduced Leco[Jo et al., 2022] and DEIR[Wan et al., 2023], two additional state-of-the-art baselines representing advanced mechanisms for episodic intrinsic rewards. These results in Table 6 demonstrate that our method consistently maintains a performance advantage in terms of effectiveness and efficiency, even when compared to advanced episodic intrinsic reward mechanisms.

Table 6: Comparative results between SI2E and advanced episodic intrinsic reward mechanisms in MiniGrid benchmark: : "average value $\pm$ standard deviation" and "average improvement"

| MiniGrid | RedBlueDoors-6x6 | | SimpleCrossingS9N1 | | KeyCorridorS3R1 | |
|---|---|---|---|---|---|---|
| Navigation | Success Rate (%) | Required Step ($K$) | Success Rate (%) | Required Step ($K$) | Success Rate (%) | Required Step ($K$) |
| Leco | $81.97 \pm 10.81$ | $817.47 \pm 137.21$ | $90.02 \pm 4.13$ | $417.59 \pm 17.63$ | $90.36 \pm 0.57$ | $520.43 \pm 10.31$ |
| DEIR | $78.32 \pm 7.21$ | $722.37 \pm 81.93$ | $91.47 \pm 8.29$ | $523.79 \pm 31.27$ | $91.81 \pm 2.13$ | $735.87 \pm 9.24$ |
| SI2E | $85.80 \pm 1.48$ | $461.90 \pm 61.53$ | $93.64 \pm 1.63$ | $139.17 \pm 27.03$ | $94.20 \pm 0.42$ | $129.06 \pm 6.11$ |
| MiniGrid | DoorKey-6x6 | | DoorKey-8x8 | | Unlock | |
| Navigation | Success Rate (%) | Required Step ($K$) | Success Rate (%) | Required Step ($K$) | Success Rate (%) | Required Step ($K$) |
| Leco | $94.37 \pm 3.41$ | $571.31 \pm 31.27$ | $92.07 \pm 19.11$ | $2168.35 \pm 293.52$ | $94.48 \pm 6.39$ | $791.40 \pm 82.39$ |
| DEIR | $94.81 \pm 5.13$ | $410.25 \pm 29.16$ | $95.41 \pm 13.27$ | $1247.58 \pm 231.42$ | $95.13 \pm 12.74$ | $531.06 \pm 131.84$ |
| SI2E | $97.04 \pm 1.52$ | $230.60 \pm 19.85$ | $98.58 \pm 3.11$ | $1090.96 \pm 125.77$ | $97.13 \pm 3.35$ | $309.14 \pm 53.71$ |

### E.2 Experiments on MetaWorld Benchmark

Figure 10 summarizes the learning curves for the DrQv2 algorithm with different exploration methods, including SI2E (ours), SE, and VCSE. These variants are denoted as DrQv2, DrQv2+SE, DrQv2+VCSE, and DrQv2+SI2E, respectively. As shown in Figure 10, SI2E consistently and significantly achieves higher success rates with fewer environmental steps across all manipulation tasks. On average, SI2E improves the final performance by $10.21\%$ and sample efficiency by $45.06\%$.

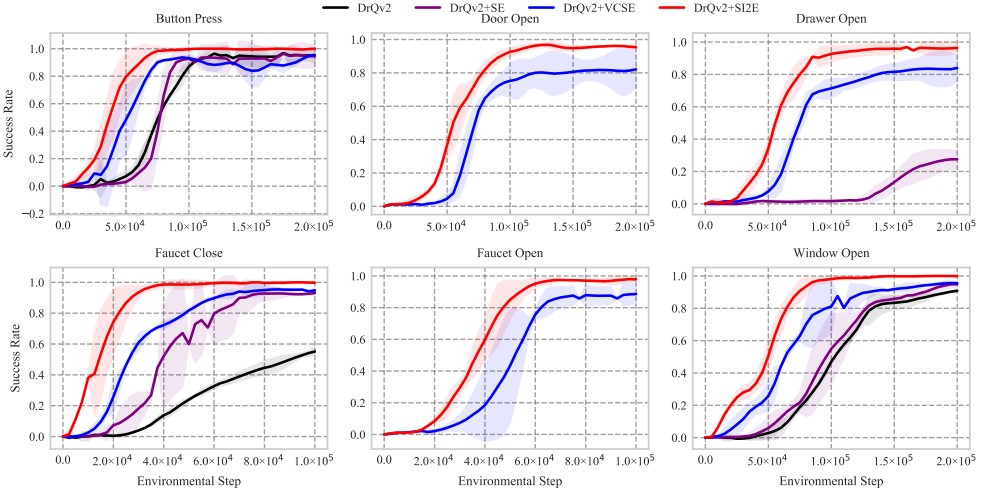

Figure 10: Learning curves for six manipulation tasks in MetaWorld, measured in terms of success rate. The solid lines represent the interquartile mean, while the shaded regions indicate the standard deviation, both calculated across 10 runs.

To provide additional validation, we conducted further experiments in the MetaWorld environment using the advanced TACO [Zheng et al., 2024] as the underlying agent. The new experimental results, as documented in Table 7, demonstrate the robustness and superior performance of our method with different underlying agents.

Table 7: Summary of success rates and required steps for the TACO agent in MetaWorld tasks: "average value $\pm$ standard deviation" and "average improvement". **Bold**: the best performance, underline: the second performance.

| MetaWorld | Button Press | | Door Open | | Drawer Open | |
|---|---|---|---|---|---|---|
| Manipulation | Success Rate (%) | Required Step ($K$) | Success Rate (%) | Required Step ($K$) | Success Rate (%) | Required Step ($K$) |
| TACO | $73.56 \pm 21.43$ | - | - | - | $33.47 \pm 2.15$ | - |
| TACO+VCSE | $\underline{91.43} \pm 5.51$ | $95.0 \pm 5.0$ | $\underline{78.59} \pm 4.22$ | - | $\underline{70.93} \pm 6.65$ | - |
| TACO+SI2E | $\mathbf{95.63} \pm 0.62$ | $\mathbf{75.0} \pm 5.0$ | $\mathbf{96.72} \pm 4.98$ | $\mathbf{115.0} \pm 10.0$ | $\mathbf{94.92} \pm 2.06$ | $\mathbf{75.0} \pm 5.0$ |
| Abs.(%) Avg. | 4.2(4.59) ↑ | 20.0(21.05) ↓ | 18.13(23.07) ↑ | - | 23.99(33.82) ↑ | - |
| MetaWorld | Faucet Close | | Faucet Open | | Window Open | |
| Manipulation | Success Rate (%) | Required Step ($K$) | Success Rate (%) | Required Step ($K$) | Success Rate (%) | Required Step ($K$) |
| TACO | $58.16 \pm 4.23$ | - | $49.48 \pm 11.60$ | - | $76.86 \pm 5.79$ | - |
| TACO+VCSE | $\underline{76.00} \pm 0.75$ | - | $\underline{80.40} \pm 3.65$ | - | $\underline{92.31} \pm 3.17$ | $\underline{120.0} \pm 5.0$ |
| TACO+SI2E | $\mathbf{93.78} \pm 1.21$ | $\mathbf{40.0} \pm 5.0$ | $\mathbf{92.09} \pm 2.26$ | $\mathbf{51.25} \pm 1.25$ | $\mathbf{97.03} \pm 1.58$ | $\mathbf{70.0} \pm 2.5$ |
| Abs.(%) Avg. | 17.78(23.39) ↑ | - | 11.69(14.54) ↑ | - | 4.72(5.11) ↑ | 50.0(41.67) ↓ |

## E.3 Experiments on DMControl Suite

For each task, we benchmark the convergence reward of the best-performing baseline as the target and track the environmental steps required by both SI2E and this baseline to reach the target. As illustrated in Figure 11, SISA demonstrates an average improvement of 35.60% in sample efficiency. This improvement is reflected in a reduction in the required environmental steps, decreasing from 31.83% to 20.5% of the total steps required to achieve the reward target.

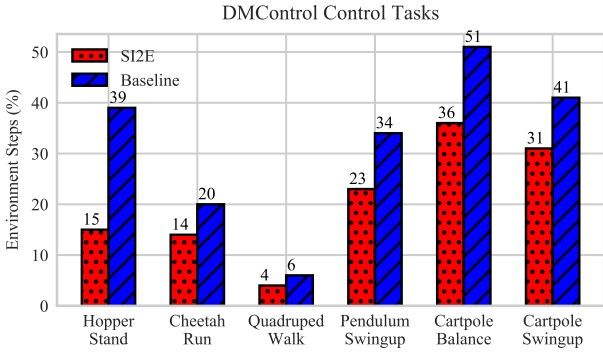

Figure 11: Comparison of sample efficiency between SI2E and the best-performing baseline in DMControl, focusing on the required environmental steps to reach the reward target, expressed as a proportion of the total $250K$ steps.

Figure 12 shows the learning curves for the DrQv2 algorithm when integrated with our SI2E framework and other exploration baselines, SE, VCSE, and MADE. The variants are identified as DrQv2, DrQv2+SE, DrQv2+VCSE, DrQv2+MADE, and DrQv2+SI2E. These results reveal that SI2E exploration significantly improves the sample efficiency of DrQv2 in both sparse reward and dense reward tasks, outperforming all other baselines. Particular in sparse reward tasks (Cartpole Balance and Cartpole Swingup), our framework successfully accelerates training and achieves higher episode rewards. This suggests that SI2E avoids exploring states that may not contribute to task resolution, thereby enhancing performance with a 5.17% improvement in final performance and a 15.28% increase in sample efficiency.

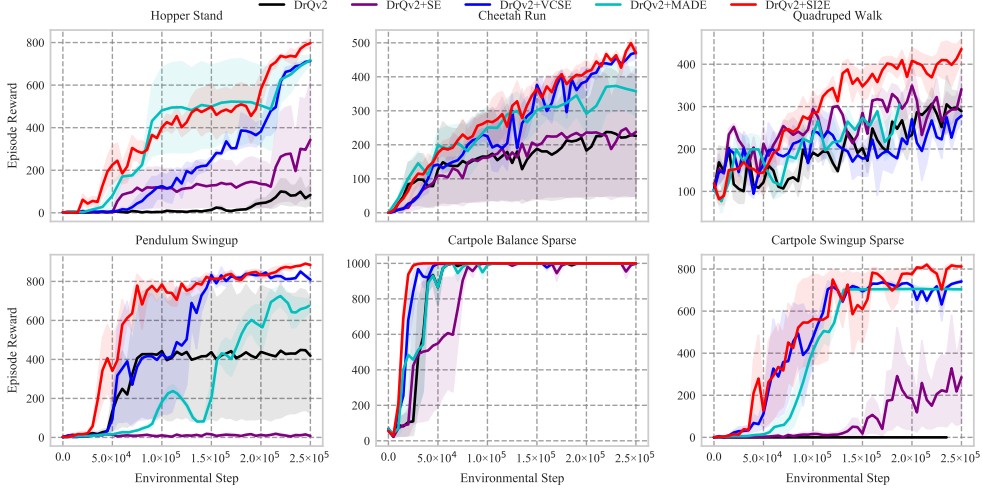

Figure 12: Learning curves for six continuous control tasks from DMControl Suite, measured in terms of episode reward. The solid lines represent the interquartile mean, while the shaded regions indicate the standard deviation, both calculated across 10 runs.

### E.4  Visualization Experiment.

To understand the learned representation through our structural mutual information principle (Section 4.1), we use t-SNE to project the original observation $O$ and state-action embedding $Z$ into 2-dimensional vectors. Figure 13a visualizes these 2-dimensional vectors and measures the Euclidean distance between temporally consecutive vectors. Our presented principle aligns all movements on the same curve and reduces their distance by an average of $36.41$, demonstrating its effectiveness for dynamics-relevant state-action embedding.

To analyze the benefits of our intrinsic reward mechanism (Section 4.2), we keep the map configuration constant in the SimpleCrossing task and display the exploration paths of various methods in Figure 13b. Our SI2E, in comparison to the SE and VCSE baselines, prevents biased exploration towards low-value areas and effectively explores high-value crucial areas, like the crossing point, with fewer environmental steps, ensuring its performance superiority.

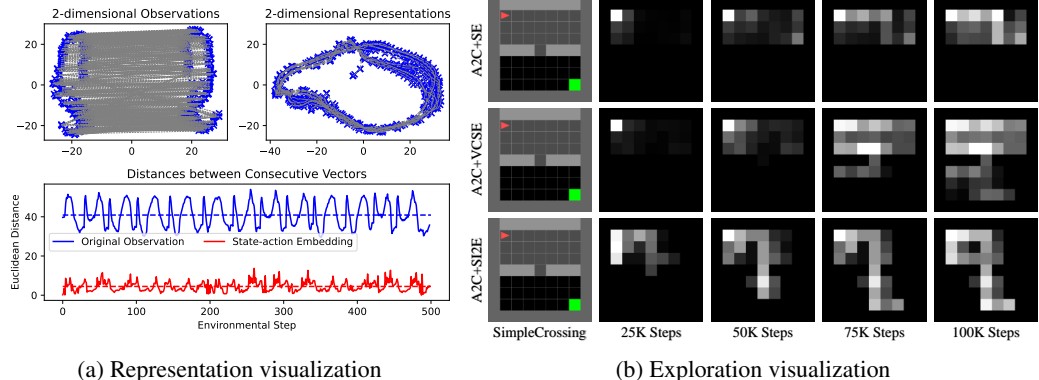

(a) Representation visualization    (b) Exploration visualization

Figure 13: Visualization results of SI2E under DMControl and Minigrid tasks. (a) Visualization of representation learning based on structural mutual information principle. (b) Visualization of agent exploration through maximizing the value-conditional structural entropy.

Moreover, we have provided the visualizations of state coverage of our SI2E framework and other baselines (SE and VCSE) in the continuous Cartpole Balance task in Figure 14. Compared with the SE and VCSE baselines, our SI2E framework achieves a more efficient exploration strategy by minimizing occurrences in extreme positions and angles, resulting in a higher density in key areas.

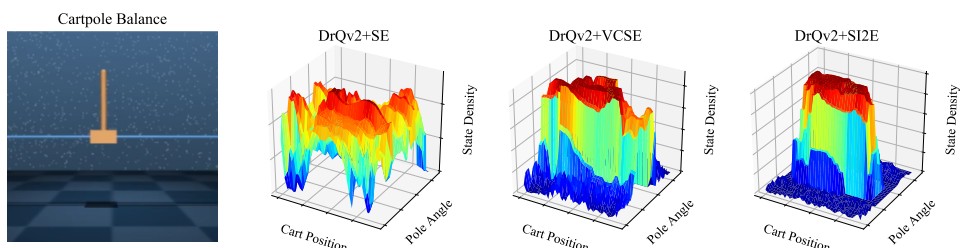

Figure 14: Visualization of agent exploration in the CartPole Balance task. Heat maps illustrate the final state densities for cart position and pole angle of the learned policies: (b)DrQv2+SE, (c) DrQv2+VCSE, and (d) DrQv2+SI2E.

## E.5  Ablation Studies

To investigate the influence of parameters $\beta$ and $n$ on our framework's performance, we incrementally adjust these parameters across two distinct tasks: DMControl tasks Hopper Stand and Pendulum Swingup. We meticulously document the resulting learning curves to assess the outcomes. Figure 15a illustrates that an increase in parameter $\beta$ consistently enhances performance across both tasks, substantiating the effectiveness of our exploration method. Conversely, Figure 15b demonstrates that variations in batch size $n$ yield comparable performance outcomes, particularly notable in Pendulum Swingup task, thereby confirming the SI2E's stability amidst variations in batch size.

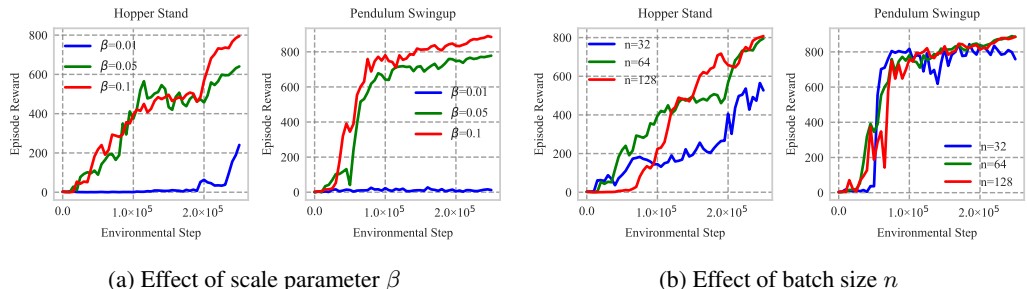

(a) Effect of scale parameter $\beta$                     (b) Effect of batch size $n$

Figure 15: Learning curves of SI2E with varied $\beta$ and $n$ values on Hopper Stand and Pendulum Swingup tasks. (a) shows the effect of the scale parameter $\beta$ on episode reward. (b) shows the effect of the batch size $n$ on episode reward. The solid line represents the interquartile mean across 10 runs.

