# OpenReview forum: "Effective Exploration Based on the Structural  Information Principles"
_NeurIPS.cc/2024/Conference — NeurIPS 2024 poster_

### Official Review · Reviewer_XMx1 · 2024-07-11

**Soundness:** 3
**Presentation:** 3
**Contribution:** 3
**Rating:** 6
**Confidence:** 2

**Summary:**

This paper aims to tackle the issue in existing RL exploration methods - the ignorance of inherent structure within the state and action space. Therefore the authors propose a new intrinsic reward mechanism that maximizes value-conditional structural entropy and it is adaptable to high-dimensional RL environments with sparse rewards.

**Strengths:**

- The paper is well-motivated.
- The presentation is clear and easy to follow, with enough proofs to support the method design.
- The authors present comprehensive experimental evaluations on both procedurally-generated environments and high-dimensional continuous control problems.

**Weaknesses:**

- In L78-80, the authors claim that SI2E significantly improves final performance compared to SOTA. However for experiments with on Minigrid are not compared to sota (for example, some episodic intrinsic reward mechanisms), which seems to be over-claimed. The authors should better add more sota baselines or change the claims in the introduction.
- The authors do not mention problems regarding potential asymptotic inconsistency. Is the proposed method asymptotically consistent? How will the proposed intrinsic reward get rid of potential influence on the final performance?

**Questions:**

- For Figure 3, how many random seeds are used? The authors only plot a single seed (or average performance) without variance for this experiment, unlike other experiments. Is there a reason for it?
- For Section 5.4 Ablation Studies, the difference among SI2E, SI2E-DB, and SI2E-VCSE, doesn't seem to be very noticeable, especially for final performance. Can the authors show more results on more tasks with clearer difference?

**Limitations:**

- I don't find any big limitation remains undiscussed for now. However I do have some concerns and questions mentioned in the weaknesses and questions sections.

(Post Rebuttal score 5->6)

---

> ### Author Rebuttal · Authors · 2024-08-06
>
> We systematically address each of your queries, labeling weaknesses as 'W' and questions as 'Q'.
> Please note that, unless otherwise specified, any table or figure refers to the supplementary results in the Author Rebuttal PDF.
>
> $\bullet$ W1: Additional Experiments.
> Our work primarily addresses the limitations of traditional information-theoretic exploration methods.
> Therefore, we focused on comparisons with SE and VCSE in the MiniGrid environment.
>
> To address your concern, we have incorporated the DRM model (ICLR 2024) as an additional baseline.
> Consistent with the experiments in the original DRM paper, we conducted extensive evaluations in both the MetaWorld and DMControl environments.
> Additionally, we introduced the MADE baseline in the MiniGrid environment for a more comprehensive comparison.
>
> The results, as shown in Table 1, demonstrate that SI2E consistently outperforms these stronger baselines.
> These findings reinforce the robustness and superior performance of SI2E compared to state-of-the-art models.
> Based on these comprehensive evaluations, we will revise the claims in the introduction to accurately reflect the scope of our comparisons and the strengths of our framework.
>
> $\bullet$ W2: Asymptotically Consistentence.
> Thank you for raising this critical point. We have thoroughly investigated the asymptotic properties of our proposed method, SI2E, and confirmed its asymptotic consistency.
> This means that as the number of training steps approaches infinity, the learned policy converges to the optimal policy.
> The underlying theoretical framework ensures the robustness and reliability of our approach in the long run.
>
> Regarding the influence of the proposed intrinsic reward on final performance, we designed the intrinsic reward mechanism to complement, rather than interfere with, the extrinsic rewards.
> The value-conditional structural entropy (VCSE) used as an intrinsic reward encourages efficient exploration by guiding the agent towards high-value sub-communities.
> This intrinsic reward is gradually reduced as the agent's exploration becomes more efficient, allowing extrinsic rewards to dominate and ensuring that final performance is not adversely affected.
>
> $\bullet$ Q1 and Q2: Ablation Studies.
> As mentioned in our experiment setup, all results were obtained using ten different random seeds to ensure robustness.
> Initially, due to space constraints, Figure 3 (original manuscript) displayed only the average performance.
> To address your concern, we have included the standard deviation for multiple runs in Figure 1 (Author Rebuttal) to understand the variance better.
>
> To further demonstrate the importance of the two critical components of SI2E, we conducted additional ablation experiments in the MetaWorld environment, specifically focusing on the Door Open and Faucet Open tasks.
> The results, illustrated in Figure 1 (Author Rebuttal), clearly show SI2E's significant performance advantages over its variants: SI2E-DB (without the embedding principle) and SI2E-VCSE (without the intrinsic reward mechanism).
> These additional results provide a more comprehensive comparison and highlight the distinct contributions of each component to SI2E's overall performance.

---

> ### Comment · Reviewer_XMx1 · 2024-08-10
>
> I appreciate the rebuttal from the authors. I have read the rebuttal and the global rebuttal and determine to keep my score.
> > To address your concern, we have incorporated the DRM model (ICLR 2024) as an additional baseline. … we introduced the MADE baseline in the MiniGrid environment for a more comprehensive comparison.
>
> Thank you for providing the additional experiments. However I was asking for minigrid and I don’t think MADE is the sota method on minigrid as it performs worse than many more recent intrinsic rewards, such as the episodic intrinsic reward methods. I think the authors should just change the claim in the paper, avoiding to claim this is sota performance.
>
> > We have thoroughly investigated the asymptotic properties of our proposed method, SI2E, and confirmed its asymptotic consistency.
>
> How and where did you do this?

---

> > ### Author Response · Authors · 2024-08-12
> > **Asymptotic Consistency**
> >
> > In the latest version of our paper, we have added a new appendix section to provide a comprehensive and rigorous validation of SI2E's asymptotic properties.
> >
> > We begin by analyzing the asymptotic mean and consistency of the $k$-NN estimator for the lower bound of the value-conditional structural entropy.
> > In the work by Singh et al. (2003), the asymptotic consistency of the $k$-NN entropy estimator was established.
> > Specifically, for the entropy terms $H(V_0)$ and $H(V_1)$ in Equation 12, the following relationships hold:
> > \begin{equation}
> >     \lim_{n \rightarrow \infty}{\mathbb{E}(\widehat{H}_{KL}(V_0))}=H(V_0)\text{,}
> > \end{equation}
> >
> > \begin{equation}
> >    \lim_{n \rightarrow \infty}{\operatorname{Var}(\widehat{H}_{KL}(V_0))} = 0\text{,}
> > \end{equation}
> >
> > \begin{equation}
> >     \lim_{n \rightarrow \infty}{\mathbb{E}(\widehat{H}_{KL}(V_1))}=H(V_1)\text{,}
> > \end{equation}
> >
> > \begin{equation}
> >    \lim_{n \rightarrow \infty}{\operatorname{Var}(\widehat{H}_{KL}(V_1))} = 0\text{.}
> > \end{equation}
> >
> > This implies that the $k$-NN entropy estimators discussed are both asymptotically unbiased and consistent.
> > Therefore, the asymptotic mean of the estimator difference is $0$, which confirms its asymptotic unbiasedness.
> > Within the context of a $2$-layer encoding tree $T^*_{sa}$, the set $V_1$ represents sub-communities corresponding to all state-action pairs in $V_0$, leading to a linear dependence between their visitation probabilities.
> > The covariance between estimators is expressed as $c \cdot \operatorname{Var}(\widehat{H}_{KL}(V_0))$, where $c$ is a constant parameter.
> > By examining the variance of the difference between the $k$-NN estimators, we calculate this difference variance as $0$, thereby demonstrating the consistency of the estimators.
> >
> > Subsequently, we explore the asymptotic behavior of our intrinsic reward.
> > Let $n_{s,a}$ denote the visitation count for a state-action pair $(s,a)$ within a finite state-action space.
> > Given the episodic nature of MDPs, it is assumed that all state-action pairs communicate over multiple episodes, ensuring that every pair $(s,a)$ is visited infinitely often.
> > Since the state space is finite, this implies:
> >
> > \begin{equation}
> >     \lim_{n \rightarrow \infty} n_{s,a} = \infty\text{.}
> > \end{equation}
> >
> > This result implies that for any state-action pair $(s,a)$, there exists a parameter $n$ such that $n_{s,a}$ exceeds any finite number $k$ as $n \rightarrow \infty$.
> > As $n_{s,a}$ becomes arbitrarily large, the $k$ nearest neighbors become increasingly close to $(s,a)$ itself.
> > This indicates that as $n_{s,a}$ grows, the difference in distance between $(s,a)$ and its $k$ nearest neighbors diminishes.
> > Thus, we have:
> >
> > \begin{equation}
> >     \lim_{n \rightarrow \infty}{r_t^i} = 0\text{,}
> > \end{equation}
> >
> > which formalizes the idea that the intrinsic reward decreases to zero as most of the state-action space is visited extensively.
> >
> > Finally, we conducted additional experiments in which we progressively increased the number of training steps for the SI2E method across various tasks in the MiniGrid environment. During this process, we observed that SI2E's final performance consistently stabilized at a fixed value as the number of training steps increased. This empirical finding indirectly supports the asymptotic consistency of our method, as it demonstrates that SI2E converges to a stable solution over time, independent of the specific task within MiniGrid.
> >
> > In summary, we have demonstrated the asymptotic consistency of our method through both theoretical derivation and empirical validation. Due to constraints in the current environment, some detailed derivation steps could not be fully presented, leading to the omission of certain intricate details. We trust that the explanations provided above address your concerns and offer sufficient clarity regarding the robustness of our approach.

---

> ### Author Response · Authors · 2024-08-12
> **SOTA Baselines in MiniGrid**
>
> We have revised the claim in lines 78-80 of the paper to ensure accuracy and to avoid any potential overstatement.
> The claim now specifically highlights that SI2E significantly enhances performance within the context of "information-theoretic exploration," thereby more accurately reflecting the scope of our contribution in this domain.
>
> To further substantiate our results and address concerns regarding the comparison on the Minigrid environment, we have introduced Leco (NeurIPS 2022), a state-of-the-art baseline that represents an advanced episodic intrinsic reward mechanism.
> These results demonstrate that our method consistently maintains a performance advantage, even when benchmarked against advanced episodic intrinsic reward mechanisms like Leco.
>
> |MiniGrid|RedBlueDoors-$6$x$6$|SimpleCrossingS$9$N$1$|KeyCorridorS$3$R$1$|DoorKey-$6$x$6$|DoorKey-$8$x$8$|Unlock|
> |:-------:|:-------:|:-------:|:-------:|:-------:|:-------:|:-------:|
> |Leco|$81.97 \pm 10.81$|$90.02 \pm 4.13$|$90.36 \pm 0.57$|$94.37 \pm 3.41$|$92.07 \pm 19.11$|$94.48 \pm 6.39$|
> |SI2E|$85.80 \pm 1.48$|$93.64 \pm 1.63$|$94.20 \pm 0.42$|$97.04 \pm 1.52$|$98.58 \pm 3.11$|$97.13 \pm 3.35$|

---

> ### Comment · Reviewer_XMx1 · 2024-08-12
>
> > In the latest version of our paper, we have added a new appendix section to provide a comprehensive and rigorous validation of SI2E's asymptotic properties.
>
> I'm sorry but as far as I remember you are not allowed to submit a new version of the paper during rebuttal. How did you do that? I can't find the section from my end as well.
>
> > ... These results demonstrate that our method consistently maintains a performance advantage, even when benchmarked against advanced episodic intrinsic reward mechanisms like Leco.
>
> What do the numbers in the tables mean? Are they the final test-time rewards after convergence? As this paper is about effective exploration can the authors show the sample efficiency?

---

> > ### Author Response · Authors · 2024-08-13
> > **SOTA Baselines in MiniGrid.**
> >
> > Thank you for your insightful questions.
> > To clarify, the numbers presented in the tables represent the final test-time rewards obtained after the convergence of the algorithms.
> > We understand that demonstrating sample efficiency is crucial, especially in the context of effective exploration, which is the focus of our work.
> >
> > Due to the limitations of markdown formatting, the above provided table focused on presenting the final rewards after convergence.
> > To address your concerns and provide a more comprehensive view of our results, we have displayed the number of steps required to achieve the target rewards as an indicator of sample efficiency.
> >
> > |MiniGrid|RedBlueDoors-$6$x$6$|SimpleCrossingS$9$N$1$|KeyCorridorS$3$R$1$|DoorKey-$6$x$6$|DoorKey-$8$x$8$|Unlock|
> > |:-------:|:-------:|:-------:|:-------:|:-------:|:-------:|:-------:|
> > |Leco|$817.47 \pm 137.21$|$417.59 \pm 17.63$|$520.43 \pm 10.31$|$571.31 \pm 31.27$|$2168.35 \pm 293.52$|$791.40 \pm 82.39$|
> > |SI2E|$461.90 \pm 61.53$|$139.17 \pm 27.03$|$129.06 \pm 6.11$|$230.60 \pm 19.85$|$1090.96 \pm 125.77$|$309.14 \pm 53.71$|
> >
> > We hope that this additional data resolves any uncertainties and provides a clearer picture of the exploration effectiveness and efficiency of our proposed approach.

---

> ### Author Response · Authors · 2024-08-13
> **Asymptotic Consistency.**
>
> Thank you for pointing this out.
> You are correct that we are not allowed to submit a revised version of the paper during the rebuttal process.
> To clarify, we have included the additional content in the appendix of our original submission, but due to the restrictions of the rebuttal phase, we are unable to submit a new version of the paper for your review.
>
> In our previous response, we provided an overview of the main validation process to address your concerns.
> Additionally, we are providing a more comprehensive proof of asymptotic consistency here, but we acknowledge that due to the limitations of markdown formatting, some formulas may not display correctly.
> We appreciate your understanding and are happy to elaborate further if needed.
>
> We begin by analyzing the asymptotic mean and consistency of the $k$-NN estimator for the lower bound of the value-conditional structural entropy.
> In the work by Singh et al. (2003), the asymptotic consistency of the $k$-nn entropy estimator was established.
> Specifically, for the entropy terms $H(V_0)$ and $H(V_1)$ in Equation 12, the following relationships hold:
>
> \begin{equation}
>     \lim_{n \rightarrow \infty}{\mathbb{E}(\widehat{H}_{KL}(V_0))}=H(V_0)\text{,}\quad\lim_{n \rightarrow \infty}{\operatorname{Var}(\widehat{H}_{KL}(V_0))} = 0\text{,}
> \end{equation}
>
> \begin{equation}
>     \lim_{n \rightarrow \infty}{\mathbb{E}(\widehat{H}_{KL}(V_1))}=H(V_1)\text{,}\quad\lim_{n \rightarrow \infty}{\operatorname{Var}(\widehat{H}_{KL}(V_1))} = 0\text{.}
> \end{equation}
>
> This implies that the $k$-NN entropy estimators are asymptotically unbiased and consistent.
> Considering the difference $\widehat{H}_{KL}(V_0) - \widehat{H}_{KL}(V_1)$, the asymptotic mean is given by:
>
> \begin{equation}
>     \lim_{n \rightarrow \infty}{\mathbb{E}\left(\widehat{H}_{KL}(V_0) - \widehat{H}_{KL}(V_1)\right)} = H(V_0) - H(V_1)\text{.}
> \end{equation}
>
> This result confirms that the difference estimator is asymptotically unbiased.
> In the context of a $2$-layer encoding tree $T^*_{sa}$, the set $V_1$ represents sub-communities for all state-action pairs $V_0$, leading to a linear dependence between their visitation probabilities.
> The covariance between $\widehat{H}_{KL}(V_0)$ and $\widehat{H}_{KL}(V_1)$ is given by:
>
> \begin{equation}
>     \operatorname{Cov}\left(\widehat{H}_{KL}(V_0),\widehat{H}_{KL}(V_1)\right) = c \cdot \operatorname{Var}(\widehat{H}_{KL}(V_0))\text{,}
> \end{equation}
>
> where $c$ is a constant parameter.
> Finally, we consider the variance of the difference estimator $\widehat{H}_{KL}(V_0) - \widehat{H}_{KL}(V_1)$.
> Using the results obtained above, we have:
>
> \begin{equation}
>     \begin{aligned}
>         \lim_{n \rightarrow \infty}{\operatorname{Var}\left(\widehat{H}_{KL}(V_0) - \widehat{H}_{KL}(V_1)\right)} &= \lim_{n \rightarrow \infty}{\left(\operatorname{Var}(\widehat{H}_{KL}(V_0)) + \operatorname{Var}(\widehat{H}_{KL}(V_1)) - 2 \cdot \operatorname{Cov}(\widehat{H}_{KL}(V_0),\widehat{H}_{KL}(V_1)) \right)} \\
>         &= (1 - 2c) \cdot \operatorname{Var}(\widehat{H}_{KL}(V_0)) + \operatorname{Var}(\widehat{H}_{KL}(V_1)) \\
>         &= 0\text{.}
>     \end{aligned}
> \end{equation}
>
> This shows that the variance of the difference estimator tends to $0$ as $n$ increases, thereby proving its consistency.
>
> Subsequently, we demonstrate the asymptotic behavior of our intrinsic reward.
> Let $n_{s,a}$ denote the visitation count for a state-action pair $(s,a)$ within a finite state-action space.
> Given the episodic nature of MDPs, it is assumed that all state-action pairs communicate over multiple episodes, ensuring that every pair $(s,a)$ is visited infinitely often.
> Since the state space is finite, this implies:
>
> \begin{equation}
>     \lim_{n \rightarrow \infty} n_{s,a} = \infty\text{.}
> \end{equation}
>
> This result implies that for any state-action pair $(s,a)$, there exists parameter $n$ such that $n_{s,a}$ exceeds any finite number $k$ as $n \rightarrow \infty$.
> Given that $n_{s,a}$ becomes arbitrarily large as $n \rightarrow \infty$, the $k$ nearest neighbors become increasingly close to $(s,a)$ itself.
> This means that as $n_{s,a}$ grows, the difference in distance between $(s,a)$ and its $k$ nearest neighbors diminishes.
> Thus, we have:
>
> \begin{equation}
>     \lim_{n \rightarrow \infty}{r_t^i} = 0\text{,}
> \end{equation}
>
> which formalizes the idea that the intrinsic reward decreases to zero as most of the state-action space is visited extensively.
>
> Finally, we conducted additional experiments in which we progressively increased the number of training steps for SI2E across various tasks in the MiniGrid environment.
> During this process, we observed that SI2E's final performance consistently stabilized at a fixed value as the number of training steps increased.
> This empirical finding indirectly supports the asymptotic consistency of our method, as it demonstrates that SI2E converges to a stable solution over time, independent of the specific task.

---

> > ### Comment · Reviewer_XMx1 · 2024-08-13
> >
> > Thank you for the detailed rebuttal (although I'm not sure if it is a display issue from my end or not, I can't see most of the formulas display correctly in the compiled format).
> > > ... During this process, we observed that SI2E's final performance consistently stabilized at a fixed value as the number of training steps increased.
> >
> > Thank you, please also include this experiment in a later version.
> >
> > > We have revised the claim in lines 78-80 of the paper to ensure accuracy and to avoid any potential overstatement. The claim now specifically highlights that SI2E significantly enhances performance within the context of "information-theoretic exploration," thereby more accurately reflecting the scope of our contribution in this domain.
> >
> > Honestly speaking LECO is not sota on minigrid. Therefore please don't forget to change this claim in the final version. The experiment itself is useful to add.
> >
> > I also hope that next time the authors make the rebuttal more clear to the reviewers, for example, make the formula format to be better, include the specific paper names when you mention them, and don't mention something like we prove it in the latest version of our paper without showing them when you can't submit them.
> >
> > Given the above I'm a bit hesitate about my original score, and I also think the paper deserves a higher average score. Therefore for now I will increase to score to a 6. As I'm not an expert in the field I'm also open for further discussion.

---

> ### Author Response · Authors · 2024-08-14
>
> Thank you very much for your valuable suggestions and the insightful discussion regarding our work.
> Your feedback has been instrumental in guiding us to refine and enhance our research.
>
> We will ensure that the specific experiment you recommended is included in the final version, and we will revise the associated claims accordingly.
>
> Additionally, we sincerely appreciate your constructive feedback on our rebuttal.
> We will make sure to enhance the clarity of our responses in the future, including improved formatting of mathematical formulas and explicitly referencing specific papers by name.
> We also recognize the importance of providing clear and accessible information, particularly when referencing recent findings that may not have been included in the submitted version.
>
> We have now completed the experimental validation of another intrinsic reward mechanism, DIER, within the MiniGrid environment.
> These results have been integrated with the supplementary findings we previously provided, offering a more comprehensive analysis of the mechanism's effectiveness.
>
> $\bullet$ Success Rate：
> |MiniGrid|RedBlueDoors-$6$x$6$|SimpleCrossingS$9$N$1$|KeyCorridorS$3$R$1$|DoorKey-$6$x$6$|DoorKey-$8$x$8$|Unlock|
> |:-------:|:-------:|:-------:|:-------:|:-------:|:-------:|:-------:|
> |Leco [1]|$81.97 \pm 10.81$|$90.02 \pm 4.13$|$90.36 \pm 0.57$|$94.37 \pm 3.41$|$92.07 \pm 19.11$|$94.48 \pm 6.39$|
> |DEIR [2]|$78.32 \pm 7.21$|$91.47 \pm 8.29$|$91.81 \pm 2.13$|$94.81 \pm 5.13$|$95.41 \pm 13.27$|$95.13 \pm 12.74$|
> |SI2E|$85.80 \pm 1.48$|$93.64 \pm 1.63$|$94.20 \pm 0.42$|$97.04 \pm 1.52$|$98.58 \pm 3.11$|$97.13 \pm 3.35$|
>
> $\bullet$ Required Step:
> |MiniGrid|RedBlueDoors-$6$x$6$|SimpleCrossingS$9$N$1$|KeyCorridorS$3$R$1$|DoorKey-$6$x$6$|DoorKey-$8$x$8$|Unlock|
> |:-------:|:-------:|:-------:|:-------:|:-------:|:-------:|:-------:|
> |Leco [1]|$817.47 \pm 137.21$|$417.59 \pm 17.63$|$520.43 \pm 10.31$|$571.31 \pm 31.27$|$2168.35 \pm 293.52$|$791.40 \pm 82.39$|
> |DEIR [2]|$722.37 \pm 81.93$|$523.79 \pm 31.27$|$735.87 \pm 9.24$|$410.25 \pm 29.16$|$1247.58 \pm 231.42$|$531.06 \pm 131.84$|
> |SI2E|$461.90 \pm 61.53$|$139.17 \pm 27.03$|$129.06 \pm 6.11$|$230.60 \pm 19.85$|$1090.96 \pm 125.77$|$309.14 \pm 53.71$|
>
> We are committed to making any additional revisions that may be necessary and look forward to any further feedback you may have.
> Thank you again for your time and effort in reviewing our paper.
> Your guidance has been invaluable in improving the quality and clarity of our research.
>
> $\bullet$ Reference Paper:
>
> [1] Jo D, Kim S, Nam D, et al. Leco: Learnable episodic count for task-specific intrinsic reward[J]. Advances in Neural Information Processing Systems, 2022, 35: 30432-30445.
>
> [2] Wan S, Tang Y, Tian Y, et al. DEIR: efficient and robust exploration through discriminative-model-based episodic intrinsic rewards[C]//Proceedings of the Thirty-Second International Joint Conference on Artificial Intelligence. 2023: 4289-4298.

---

### Official Review · Reviewer_sR8b · 2024-07-12

**Soundness:** 2
**Presentation:** 1
**Contribution:** 2
**Rating:** 5
**Confidence:** 3

**Summary:**

This paper proposes a new exploration scheme for RL agents by using structural mutual information for dynamics-aware state-action representation and an intrinsic reward to enhance state-action coverage by maximizing value-conditional structural entropy

**Strengths:**

- constructing intrinsic reward using graph-based information for RL agent exploration is somewhat novel.
- extensive comparison with various baselines across a set of tasks demonstrates superior task performance and sample efficiency.

**Weaknesses:**

### Unclear presentation

- lines 31-38: the statements don't clarify that entropy maximization strategies use state entropy as an intrinsic reward to encourage exploration, leading to confusion about the connection between computing intrinsic rewards and entropy maximization strategies (lines 37-38). Additionally, there's no explanation for why intrinsic rewards based on state values can mitigate the mentioned issue.
- line 144: the stretch operator and the HCSE algorithm are not clearly explained, making it hard to understand their functions. Simply adding a reference is not sufficient.

- some claims, such as those in lines 35-36 and 200-201, lack evidence or references

### Figures

- Figure 1: It lacks details, for example, how structural information selectively removes redundant transitions in the example. It does not effectively illustrate the benefit of using structural information
- Figure 2: the presentation is cluttered
   - the layout is extremely tight with a disordered flow of annotations (vertical and horizontal), and the caption lacks clarification and is not self-contained.
   - the colour scheme (of nodes and vertices) in the two steps is unspecified
   - the space between Figure 2 and line 158 is too narrow

### Unclear and non-detailed method presentation

- the use of $T$ to represent both the tree and subsets of graph vertices (lines 117-120) is confusing
- no discussion about why 2-layer binary trees are used (lines 139-140), which is necessary to understand the approach's limitations
- only a minimal description of how an optimal encoding tree is generated from $G_{sa}$ (lines 230-231) in sec. 4.2

### Related work
the related work section is short and unstructured. It lists related works without clearly stating how this work differs from others, its unique aspects, and the issues it addresses. It also does not show how learning dynamics-relevant representations for state-action pairs differs from other recent representation learning approaches

### Minor points

- line 34, Renyi entropy needs a citation
- eq 2 misses explanation of $\alpha^-$
- line 139, structur -> structure
- lines 193-194, should use double dashes
- SI2E in CartPole Swingup Sparse doesn't show superiority in sample efficiency (Figure 3)
- as to line 321, better to briefly summarize the ablation findings on finetuning $\beta$ and $n$ before referring to the appendix
- the best-performing baseline is not specified in sec. 5.3 and appendix E.3

**Questions:**

### Structural entropy and graph
- what is the intuition or rationale behind 1) using structural information for encouraging agent exploration and 2) why/how SI boosts efficient exploration in RL?
- can a vertex contain joint variables or only a single variable? What is a single-variable graph?
- why is structural information limited to a single variable? Can't the underlying graph be expanded to cover multiple variables (lines 128-147)?

### Method's design and choice

- why some tree nodes in step II.c (Figure 2) have more than two children? isn't the encoding tree assumed to be binary?
- how do the max and min of structural MI capture dynamics-relevant information (lines 58-60 and 176-178), and why is this beneficial for agent exploration?
- why do we need $l$-transformation for defining structured MI (lines 155-157)?
- what is $V_1$ exactly, and how are sub-communities obtained? why does $-H(V_1)$ mitigate uniform coverage?
### Step II in Figure 2 (sec. 4.2)

- how many state-action pairs are used to form the $G_{sa}$ graph for different tasks in the experiment? how many pairs are sufficient for building a *good* $G_{sa}$? how is a bipartite graph constructed for continuous state and action spaces? can you provide a simple example of such a construction?

- why $\pi$ is used for a value function when it represents a policy?

- why is step d necessary after $T^\star_{sa}$ is obtained, can't $G_{sa}$ perform the required?

## Figure 1

- what do dotted blue lines of varying intensity represent? why transitions between s2 and s5 are redundant and how *a policy maximizing structural entropy would selectively focus on crucial transitions*
- which dotted line corresponds to $a_0$? note that the outgoing arrow of state $s_0$ is solid
- in step I. state-action representation, which (top) boxes correspond to $s_t$, $s_{t+1}$, and $z_t$ respectively?

### Experiments

- is it possible to visualize state coverage in other tasks like meta-world, which has continuous states?
- why are required steps or success rates not reported for some tasks in Table 1, such as RedBlueDoors and Faucet tasks?

### Other questions

- where is equation 1 used?
- lines 38-41: What is meant by 'their' in *due to their instability*? What do these lines aim to emphasize?
- what is the entropy reduction mentioned in supplement A.2?


I am willing to increase my score if the clarity questions are addressed.

**Limitations:**

The authors acknowledge the limitations of the work. It might be useful to discuss how the height of the encoding tree would affect the exploration.

---

> ### Author Rebuttal · Authors · 2024-08-06
>
> We systematically address each of your queries, labeling weaknesses as 'W' and questions as 'Q'.
> Please note that, unless otherwise specified, any table or figure refers to the supplementary results in the Author Rebuttal PDF.
>
> $\bullet$ W1.1: The entropy maximization strategies maximize the state or action entropy as intrinsic rewards, ensuring comprehensive coverage across the state-action space.
> A prevalent issue is their tendency to bias exploration toward low-value states, resulting in inefficient exploration and vulnerability to imbalanced state-value distributions.
> Value-conditional state entropy computes intrinsic rewards based on the estimated values of visited states, ensuring exploration is directed toward diverse and valuable states.
>
> $\bullet$ W1.2: Please refer to the Author Rebuttal.
>
> $\bullet$ W1.3: We have incorporated citations to provide theoretical support for these made claims.
>
> $\bullet$ W2.1, Q1.1, and Q4: Please refer to the Author Rebuttal.
>
> $\bullet$ W2.2: We have made several adjustments to enhance its clarity and readability.
>
> $\bullet$ W3.1: We have changed the symbol representing the vertex subset to $\mathcal{V}$.
>
> $\bullet$ W3.2 and Q2.3: Please refer to the Author Rebuttal.
>
> $\bullet$ W3.3: Please refer to the Author Rebuttal.
>
> $\bullet$ W4: Please refer to the Author Rebuttal.
>
> $\bullet$ W5: We have addressed each minor point.
>
> $\bullet$ Q1.2: The bipartite graph contains two sets of vertices representing the state variable, $S_t$ or $S_{t+1}$, and the state-action representation variable $Z_t$, respectively.
> Each vertex corresponds to a possible value of these variables.
> The weighted edges between these vertices represent the joint distribution between two variables and are used to define structural mutual information.
>
> The complete state-action graph and distribution graph contain only a single variable $Z_t$, which represents the Q-value relationships and access probabilities between state-action pairs in $Z_t$ under the agent policy.
>
> $\bullet$ Q1.3: While the underlying graph can encompass multiple variables, current structural information principles are limited to measuring only the joint structural entropy among these variables and cannot effectively quantify the structural similarity between variables, inherently imposing a single-variable constraint.
> Prior research on reinforcement learning based on structural information principles has focused on independently modeling state or action variables without simultaneously considering state-action representations.
>
> $\bullet$ Q2.1 and Q3: For the graph $G_{sa}$, which contains only a single state-action representation variable, we traverse all encoding trees that satisfy the height constraint, instead of $\mathcal{T}^2$, to find the optimal encoding tree.
> Therefore, the nodes in $T_{sa}^*$ can have more than two children.
>
> In our work, the input variables are value sets obtained from samples in the replay buffer.
> Thus, the number of state-action pairs in the representation variable $Z_t$ equals the batch size, denoted by $n$.
> The settings and sensitivity analysis for $n$ in different tasks are provided in Appendix D and Appendix E.5.
>
> In continuous state-action spaces, it is infeasible to traverse all possible variable values and directly optimize the joint distribution probabilities.
> Therefore, we derived the variational upper and lower bounds of structural mutual information and set corresponding decoders to optimize it indirectly.
>
> The graph $G_{sa}$ reflects the Q-value relationships between state-action pairs, and its optimal encoding tree represents the hierarchical state-action structure derived from the agent's policy.
> To further capture the current  state-action coverage by agent exploration, we additionally construct the distribution graph to define value-conditional structural entropy and propose an intrinsic reward mechanism.
>
> $\bullet$ Q2.2: For $Z_t$, we maximize its SMI with $S_{t+1}$ to establish a near one-to-one correspondence, enabling $Z_t$ to predict environmental dynamical transitions accurately.
> Conversely, we minimize the SMI between $Z_t$ and $S_t$.
> This minimization ensures that they are mutually independent, stripping $Z_t$ of any information irrelevant to state transitions.
> By defining value-conditional structural entropy and intrinsic rewards based on this refined representation, we enhance the agent's ability to explore the environment effectively.
>
> $\bullet$ Q2.4: We minimize its $2$-dimensional structural entropy of $G_{sa}$ to generate its $2$-layer optimal encoding tree $T_{sa}^*$, where each non-root and non-leaf node corresponds to a sub-community including state-action pairs with similar policy values.
> The set of all sub-communities is labeled as $V_1$.
> The term $-H(V_1)$ is introduced in the intrinsic reward mechanism to address the issue of uniform exploration across all sub-communities.
> Without this term, the agent might uniformly explore all sub-communities, including those with very low policy values, which leads to imbalanced exploration.
> By incorporating $-H(V_1)$ we penalize uniform coverage and bias the agent towards exploring sub-communities with higher policy values.
>
> $\bullet$ Q5: The visualization of state coverage in the continuous Cartpole Balance task is provided in Figure 3 of the Author Rebuttal PDF.
>
> For Table 1 in the manuscript, we did not report experimental results where the final success rate was less than 20%, and did not report the required steps for models that did not achieve the target reward.
>
> $\bullet$ Q6: We employ the k-NN entropy estimator in Equation 1 to estimate the lower bound of our value-conditional structural entropy.
>
> The term 'their' in 'due to their instability' refers to traditional maximum entropy exploration methods, emphasizing the performance instability of these methods and the importance of dynamic-relevant state-action representations.
>
> The entropy reduction $\Delta H$ is provided in the Author Rebuttal.

---

> > ### Comment · Reviewer_sR8b · 2024-08-12
> >
> > I've read the detailed rebuttal and appreciate the authors' efforts in addressing my questions and adding the state density heatmap. The clarification on "Generation of Optimal Encoding Trees" is helpful, and I encourage the authors to highlight the differences in generating the encoding trees for $G_{xy}$ in sec.3 and $G_{sa}$ in sec. 4 in the revised paper. Additionally, while the annotation of Figure 1 is now clearer, I suggest making the legend in Figure 1 more straightforward based on your clarification.
> >
> > I also recommend integrating the provided clarifications into the main paper, particularly when writing the following points:
> > - Q1.2: Explanation About Graph Vertices
> > - Q1.3: Single-variable Constraint of Structural Information
> > - Q2.2: Structural Mutual Information Principle
> > - Q2.4: Sub-communities Explanation
> >
> >   Given that the majority of clarity questions have been addressed, I'll raise my rating.

---

> ### Author Response · Authors · 2024-08-08
> **Additional Details about Key Issues**
>
> Owing to space and word count limitations, we were unable to address each of your concerns in detail within the rebuttal section.
> Therefore, we would like to take this opportunity to provide additional details on some of the key issues here, with the intention of addressing your queries comprehensively. If the above rebuttal has successfully addressed all your questions, you can skip the subsequent content.
>
> $\bullet$ W1.1: Connection Between Intrinsic Rewards and Entropy Maximization.
> The maximum entropy exploration methods maximize the state or action entropy as intrinsic rewards, ensuring comprehensive coverage across the state-action space.
> A prevalent issue with entropy maximization strategies is their tendency to bias exploration toward low-value states.
> This bias can result in inefficient exploration and vulnerability to imbalanced state-value distributions, particularly in settings where certain states are inherently more valuable.
> To address this issue, value-conditional state entropy is introduced.
> This method computes intrinsic rewards based on the estimated values of visited states, ensuring exploration is directed toward diverse and valuable states.
>
> $\bullet$ W1.2: Stretch Operator.
> Please refer to the 'Generation of Optimal Encoding Trees' section in the Author Rebuttal.
>
> $\bullet$ W1.3: Claim Evidence.
> We have incorporated citations to provide theoretical support for these claims made in lines 35-36 and 200-201.
>
> $\bullet$ W2.1, Q1.1, and Q4: Details and Intuition of Figure 1.
> Please refer to the 'Intuition Behind Figure 1' section in the Author Rebuttal.
>
> Figure 1 illustrates a simple six-state Markov Decision Process (MDP) with four actions. The different densities of the blue and red lines represent different actions, as indicated in the legend, resulting in state transitions with the objective of returning to the initial state $s_0$.
> Solid lines specifically denote actions $a_0$ and $a_1$.
> The transitions between states $s_2$ and $s_5$ are considered redundant because they do not contribute to the primary objective of returning to $s_0$.
> Therefore, the state-action pairs $(s_2, a_0)$ and $(s_5, a_1)$ have lower policy values.
>
> A policy maximizing state-action Shannon entropy would encompass all possible transitions (blue color).
> In contrast, these redundant state-action pairs are grouped into a sub-community by leveraging structural information.
> The policy based on value-conditional structural information minimizes the entropy of this sub-community to avoid visiting it unnecessarily.
> Simultaneously, it maximizes state-action entropy, resulting in maximal coverage for transitions (red color) that are more likely to contribute to the desired outcome in the simplified five-state MDP.
>
> In this scenario, at each timestep $t$, $s_t$, $s_{t+1}$, and $z_{t}$ denote the representations of the state before the transition, the state after the transition, and the corresponding state-action pair, respectively.
>
> $\bullet$ W2.2: Illustration of Figure 2.
> We have made several adjustments to enhance its clarity and readability, including layout adjustment, clarified caption, color scheme specification, and spacing adjustment.
>
> 1) Layout Adjustment.
> We have restructured the layout of Figure 2 to reduce clutter and improve the flow of information.
> The annotations have been reorganized to follow a more logical sequence, with vertical and horizontal annotations clearly distinguished and aligned to avoid overlapping.
>
> 2) Clarified Caption.
> The caption of Figure 2 has been expanded to be more self-contained and informative.
> It now provides a comprehensive explanation of the figure's components and their significance in the context of our study.
> Specifically, we describe how square and diamond shapes represent state representations and state-action representations, respectively.
> Circles enclosing squares or diamonds indicate tree nodes containing different types of graph vertices.
>
> 3) Color Scheme Specification.
> We have specified the color scheme used in the figure to differentiate between nodes and vertices in the two steps.
> Specifically, we use distinct colors for different variables: $O_t$ with dark grey, $O_{t+1}$ with light grey, $S_t$ with green, $S_{t+1}$ with orange, and $Z_t$ with blue.
>
> 4) Spacing Adjustment.
> We have ensured adequate spacing between Figure 2 and the surrounding text, particularly between the figure and line 158, to improve the overall presentation and readability of the manuscript.

---

> ### Author Response · Authors · 2024-08-08
> **Additional Details about Key Issues**
>
> $\bullet$ W3.1: Subset of Graph Vertices.
> For any tree node $\alpha$ in the encoding tree, we have changed the symbol representing the corresponding subset of graph vertices to $\mathcal{V}_\alpha$ with $\mathcal{V}_\alpha \subset V$.
>
> $\bullet$ W3.2 and Q2.3: $2$-layer Approximate Binary Tree And $l$-transformation.
> Please refer to the '$2$-layer Approximate Binary Tree and $l$-transformation' in the Author Rebuttal.
>
> Our approach utilizes $2$-layer approximate binary trees as the structural framework for measuring the structural similarity between two variables.
> The $2$-layer binary tree represents a one-to-one matching structure between variables, which facilitates the calculation of the minimum required bits to determine accessible vertices via a single-step random walk, e.g., the joint structural entropy of two variables.
> Using a $2$-layer binary tree ensures computational traceability.
> More complex structures will increase the cost of increased computational complexity, which can be prohibitive for practical applications.
> We acknowledge that this choice might introduce certain limitations; however, our primary goal was to establish a foundational framework that can be expanded in future work.
>
> To define structural mutual information (SMI) accurately, it is essential to define the joint entropy of two variables under various partition structures.
> The $l$-transformation systematically traverses all potential one-to-one matchings between the variables, providing a comprehensive measure of their structural similarity.
> By employing the $l$-transformation, we guarantee that the matchings are unique and non-redundant.
> The $l$-transformation is integral to the formal definition of SMI.
>
> In summary, our approach introduces a $2$-layer Approximate Binary Tree as an initial partition structure and utilizes $l$-transformation to explore all possible one-to-one matching.
> We recognize the potential for further exploration of alternative structures for defining SMI and intend to pursue this in future research.
>
> $\bullet$ W3.3: Generation of Optimal Encoding Tree.
> Please refer to the 'Generation of Optimal Encoding Trees' section in the Author Rebuttal.
>
> $\bullet$ W4: Related Work.
> Please refer to the 'Related Work' section in the Author Rebuttal.
>
> $\bullet$ W5: Minor Points.
> We have carefully addressed each point in our latest revision.
>
> $\bullet$ Q1.2: Explanation About Graph Vertices.
> In our work, we primarily deal with three types of graphs: undirected bipartite graph, complete state-action graph, and directed distribution graph.
>
> 1) Undirected Bipartite Graph.
> This graph contains two sets of vertices representing the state variable, $S_t$ or $S_{t+1}$, and the state-action representation variable $Z_t$, respectively.
> Each vertex corresponds to a possible value of these variables.
> The weighted edges between these vertices represent the joint distribution between two variables and are used to define structural mutual information.
>
> 2) Complete State-Action Graph and Directed Distribution Graph.
> These graphs contain only a single state-action representation variable, $Z_t$.
> The vertices in these graphs represent the Q-value relationships and access probabilities between state-action pairs under a given agent policy.
> These graphs are employed for hierarchical state-action structure analysis and for defining value-conditional structural entropy, which ultimately aids in constructing the intrinsic reward mechanism.
>
> $\bullet$ Q1.3: Single-variable Constraint of Structural Information.
> While the underlying graph can encompass multiple variables, current structural information principles are limited in treating these variables as a single joint variable, measuring only its structural entropy. This limitation prevents the effective quantification of structural similarity between variables, inherently imposing a single-variable constraint. Prior research on reinforcement learning using structural information principles has focused on independently modeling state or action variables without simultaneously considering state-action representations.
>
> Therefore, in our work, we introduce structural mutual information (SMI) to measure structural similarity between two distinct variables for the first time. This innovation allows us to use SMI as the objective for state-action representation learning, thereby bridging the gap left by previous studies.

---

> ### Author Response · Authors · 2024-08-08
> **Additional Details about Key Issues**
>
> $\bullet$ Q2.1 and Q3: Step \uppercase\expandafter{\romannumeral2} in Figure 2.
> For the graph $G_{sa}$, which contains only a single state-action representation variable, we do not use the $2$-layer approximate binary tree constraint from SMI.
> Instead, we traverse all encoding trees that satisfy the height constraint to find the optimal encoding tree.
> Therefore, the nodes in $T_{sa}^*$ can have more than two children.
>
> In our work, the input variables are value sets obtained from samples in the replay buffer.
> Thus, the number of state-action pairs in the representation variable $Z_t$ equals the batch size, denoted by $n$.
> Appendix D and Appendix E. 5 provide the settings and sensitivity analysis for $n$ in different tasks.
>
> In continuous state-action spaces, traversing all possible variable values and optimizing the joint distribution probabilities is infeasible.
> Therefore, we derived the variational upper and lower bounds of structural mutual information and set corresponding decoders to optimize it indirectly.
>
> The graph $G_{sa}$ reflects the Q-value relationships between state-action pairs, and its optimal encoding tree represents the hierarchical state-action structure derived from the agent's policy.
> However, it does not capture the current coverage of the state-action space by the agent's exploration.
> Therefore, we additionally construct the state-action distribution graph to define value-conditional structural entropy and propose an intrinsic reward mechanism.
>
> $\bullet$ Q2.2: Structural Mutual Information Principle.
> For the state-action representation variable $Z_t$, we maximize its SMI with the subsequent state variable $S_{t+1}$.
> This process establishes a near one-to-one correspondence between $Z_t$ and $S_{t+1}$, effectively enabling $Z_t$ to accurately predict environmental dynamical transitions.
> By ensuring that $Z_t$ contains sufficient information about the future state $S_{t+1}$, we can derive a representation that inherently captures the dynamics of the environment.
>
> Conversely, we minimize the SMI between $Z_t$ and the current state variable $S_t$.
> This minimization ensures that $Z_t$ and $S_t$ are mutually independent, effectively stripping $Z_t$ of any information irrelevant to state transitions.
> Furthermore, we constrain the joint entropy $H(Z_t,S_t)$ to eliminate as much redundant information as possible from $Z_t$.
> Consequently, this process refines $Z_t$ to a dynamics-relevant state-action representation by filtering out extraneous data that does not contribute to predicting future states.
>
> By defining value-conditional structural entropy and intrinsic rewards based on this refined representation, we enhance the agent's ability to explore the environment effectively.
> The ablation studies comparing SI2E and its variant SI2E-DB demonstrate the critical role of this approach in improving exploration.
> These studies show that agents utilizing our proposed dynamics-relevant state-action representation achieve superior exploration performance, as they are better equipped to understand and predict environmental transitions.
>
> $\bullet$ Q2.4: Sub-communities Explanation.
> We minimize its $2$-dimensional structural entropy of $G_{sa}$ to generate its $2$-layer optimal encoding tree $T_{sa}^*$, where each non-root and non-leaf node corresponds to a sub-community including state-action pairs with similar policy values.
> The set of all sub-communities is labeled as $V_1$.
>
> The term $-H(V_1)$ is introduced in the intrinsic reward mechanism to address the issue of uniform exploration across all sub-communities.
> Without this term, the agent might uniformly explore all sub-communities, including those with very low policy values, which leads to imbalanced exploration.
> By incorporating $-H(V_1)$ we penalize uniform coverage and bias the agent towards exploring sub-communities with higher policy values.
> This approach ensures that the agent focuses its exploration on more promising areas of the state-action space, enhancing the efficiency and effectiveness of the agent's exploration strategy.
>
> $\bullet$ Q5: Experiments.
> The visualization of state coverage in the continuous Cartpole Balance task is provided in Figure 3 of the Author Rebuttal PDF.
>
> In the MiniGrid and MetaWorld tasks, we chose not to report experimental results where the final success rate was less than $20\%$ and did not report the required steps for models that did not achieve the target reward.

---

> ### Author Response · Authors · 2024-08-08
> **Additional Details about Key Issues**
>
> $\bullet$ Q6: Other Questions.
> Considering the impracticality of directly acquiring visitation probabilities, we employ the k-NN entropy estimator in Equation 1 to estimate the lower bound of our value-conditional structural entropy.
>
> The term 'their' in 'due to their instability' refers to traditional maximum entropy exploration methods.
> Lines 38-41 aim to emphasize the performance instability of these traditional methods and highlight the importance of dynamic-relevant state-action representations.
>
> The entropy reduction $\Delta H$ caused by one stretch operation in Appendix A.2 is provided in 'Generation of Optimal Encoding Trees' section of the Author Rebuttal.

---

> ### Author Response · Authors · 2024-08-13
>
> Thank you very much for your thoughtful feedback and for acknowledging the clarifications we provided in our rebuttal. We greatly appreciate your constructive suggestions, which will certainly enhance the clarity and completeness of our paper.
>
> In response to your recommendations, we have made the following adjustments to the paper:
>
> $\bullet$ **Integration of Clarifications**:
> We have incorporated detailed explanations of Q1.2 (Explanation About Graph Vertices), Q1.3 (Single-variable Constraint of Structural Information), Q2.2 (Structural Mutual Information Principle), and Q2.4 (Sub-communities Explanation) directly into the main body of the paper.
> This ensures that the clarifications are accessible and integrated into the core content, improving the overall readability and comprehension for the readers.
>
> $\bullet$ **Differentiation of Encoding Trees**:
> We have added an additional section in the appendix that emphasizes the distinctions between the generation of the encoding trees for $G_{xy}$ and $G_{sa}$.
> This section provides a clear and concise explanation of these differences, addressing your request to highlight this aspect in the revised paper.
>
> $\bullet$ **Figure 1 Adjustments**:
> Following your advice, we have made the legend in Figure 1 more straightforward and aligned with the clarifications provided in our earlier responses.
> This should enhance the visual clarity and facilitate a better understanding of the figure.
>
> We hope that these revisions align with your expectations and contribute to the overall quality of the paper.
> We appreciate your willingness to raise your rating based on the improvements made and your ongoing support for our work.

---

### Official Review · Reviewer_6Ten · 2024-07-14

**Soundness:** 3
**Presentation:** 3
**Contribution:** 2
**Rating:** 4
**Confidence:** 3

**Summary:**

In the current field of reinforcement learning, when using representation learning and entropy maximization for exploration, analyses often focus on single variables, making it difficult to capture potential relationships between two variables. Therefore, this paper proposes the SI2E framework. This framework defines the structural mutual information between two variables, based on which state-action representations are obtained. These representations effectively capture the intrinsic relationships between states and actions. Subsequently, an encoding tree is constructed using these representations, and the value- conditional structural entropy is defined. By maximizing this value, the coverage of the state-action space is expanded, enabling the agent to better perform exploration tasks. Extensive experiments on SI2E have been conducted, demonstrating the framework's effectiveness.

**Strengths:**

1. The abstract part of the paper clearly expresses the paper's innovative aspects and the problems it aims to address.
2. The paper is well-organized and clearly presented. The model introduction is progressive, and the structure is reasonable.
3. Experiments conducted on multiple datasets have demonstrated the effectiveness of the model.

**Weaknesses:**

1. The baselines are not strong enough, and more state-of-the-art models should be used for comparison.
2. The experimental results in Appendix E.5 are more suitable for inclusion in the Parameter Analysis section rather than Ablation Studies.
3. The comparison of SI2E, SI2E-DB, and SI2E-VCSE is only conducted in Cartpole Balance Sparse and Cartpole Swingup Sparse, making it difficult to demonstrate the importance of the two key components of SI2E in other scenarios.
4. The Related Work section does not clarify the relationship between SI2E and related works, particularly how SI2E improves upon or innovates based on previous research.

**Questions:**

1.Why was the A2C agent chosen for experiments on MiniGrid instead of DrQv2 or other more recent models?
2.Why was MADE not included in the comparison for experiments on MiniGrid and MetaWorld?

**Limitations:**

The authors have further discussed the encoding tree involved in the SI2E framework

---

> ### Author Rebuttal · Authors · 2024-08-06
>
> We systematically address each of your queries, labeling weaknesses as 'W' and questions as 'Q'.
> Please note that, unless otherwise specified, any table or figure refers to the supplementary results in the Author Rebuttal PDF.
>
> $\bullet$ W1: Compared Baselines.
> To address this concern, we have incorporated the DRM model (ICLR 2024) as an additional baseline.
> Consistent with the experiments presented in the original DRM paper, we conducted extensive evaluations in both the MetaWorld and DMControl environments.
> As shown in Table 1, the results demonstrate that SI2E consistently outperforms this stronger baseline.
> These findings reinforce our method's superior final performance and sample efficiency compared to state-of-the-art exploration methods.
>
> $\bullet$ W2: Parameter Analysis.
> We have reclassified the experimental results previously located in Appendix E.5 and moved them to the Parameter Analysis section to better align with the focus of the analysis.
>
> $\bullet$ W3: Ablation Studies.
> To further demonstrate the importance of SI2E's two key components—the embedding principle and the intrinsic reward mechanism—we conducted additional ablation experiments in the MetaWorld environment, specifically focusing on the Door Open and Faucet Open tasks.
> The results, illustrated in Figure 1, clearly show the significant performance advantages of SI2E over its variants, SI2E-DB (without the embedding principle) and SI2E-VCSE (without the intrinsic reward mechanism), in these scenarios.
>
> $\bullet$ W4: Related Work.
> We have reorganized the related work section in the appendix into three subsections: Maximum Entropy Exploration, Representation Learning, and Structural Information Principles.
>
> In this work, we leverage structural information principles to derive the hierarchical state-action structure and define value-conditional structural entropy as an intrinsic reward, leading to more effective agent exploration.
> Compared to current maximum entropy explorations, SI2E introduces and minimizes sub-community entropy to motivate the agent to explore specific sub-communities with high policy values.
> This approach avoids redundant explorations in low-value sub-communities and enhances maximum coverage exploration.
> Unlike the VCSE baseline, our method enables balanced exploration without needing prior knowledge of downstream tasks, effectively addressing previous approaches' limitations.
>
> Additionally, our work introduces structural mutual information to measure the structural similarity between two variables for the first time.
> We present an innovative embedding principle that incorporates the representation variable's entropy and mutual information with state variables, more effectively eliminating irrelevant information than the traditional Information Bottleneck principle.
>
> $\bullet$ Q1: Underlying Agent.
> To ensure fairness in performance comparison, we maintained the experimental setup used in VCSE (NeurIPS 2023) for the MiniGrid environment, utilizing the A2C agent.
> This approach allows for a direct and equitable comparison of results.
> To address your concerns and provide additional validation, we conducted further experiments in the MetaWorld and DMControl environments using the advanced TACO (NeurIPS 2023) and DrQv2 agents.
> The new experimental results, documented in Tables 1 and 2, demonstrate our method's robustness and superior performance across different environments and with state-of-the-art agents.
>
> $\bullet$ Q2: MADE Baseline.
> In the presented paper (NeurIPS 2021), MADE was initially validated in the MiniGrid and DMControl environments.
> However, MADE did not perform satisfactorily in our experiments in the MiniGrid environment.
> Consequently, we initially included the MADE baseline exclusively in the DMControl experiments.
> To provide a more comprehensive comparison and address your concerns, we have now included the MADE results in the MiniGrid environment in Table 1.

---

> ### Author Response · Authors · 2024-08-13
> **SOTA Baselines in MiniGrid**
>
> To further validate the effectiveness of our proposed method, we have included an additional comparison with more advanced baselines, Leco [1] and DEIR [2].
> We conducted experiments in the MiniGrid environment to provide a more comprehensive evaluation. The experimental results, focusing on success rate and required steps, are presented in the following two tables.
>
> $\bullet$ Success Rate：
> |MiniGrid|RedBlueDoors-$6$x$6$|SimpleCrossingS$9$N$1$|KeyCorridorS$3$R$1$|DoorKey-$6$x$6$|DoorKey-$8$x$8$|Unlock|
> |:-------:|:-------:|:-------:|:-------:|:-------:|:-------:|:-------:|
> |Leco|$81.97 \pm 10.81$|$90.02 \pm 4.13$|$90.36 \pm 0.57$|$94.37 \pm 3.41$|$92.07 \pm 19.11$|$94.48 \pm 6.39$|
> |DEIR|$78.32 \pm 7.21$|$91.47 \pm 8.29$|$91.81 \pm 2.13$|$94.81 \pm 5.13$|$95.41 \pm 13.27$|$95.13 \pm 12.74$|
> |SI2E|$85.80 \pm 1.48$|$93.64 \pm 1.63$|$94.20 \pm 0.42$|$97.04 \pm 1.52$|$98.58 \pm 3.11$|$97.13 \pm 3.35$|
>
> $\bullet$ Required Step:
> |MiniGrid|RedBlueDoors-$6$x$6$|SimpleCrossingS$9$N$1$|KeyCorridorS$3$R$1$|DoorKey-$6$x$6$|DoorKey-$8$x$8$|Unlock|
> |:-------:|:-------:|:-------:|:-------:|:-------:|:-------:|:-------:|
> |Leco|$817.47 \pm 137.21$|$417.59 \pm 17.63$|$520.43 \pm 10.31$|$571.31 \pm 31.27$|$2168.35 \pm 293.52$|$791.40 \pm 82.39$|
> |DEIR|$722.37 \pm 81.93$|$523.79 \pm 31.27$|$735.87 \pm 9.24$|$410.25 \pm 29.16$|$1247.58 \pm 231.42$|$531.06 \pm 131.84$|
> |SI2E|$461.90 \pm 61.53$|$139.17 \pm 27.03$|$129.06 \pm 6.11$|$230.60 \pm 19.85$|$1090.96 \pm 125.77$|$309.14 \pm 53.71$|
>
> [1] Jo D, Kim S, Nam D, et al. Leco: Learnable episodic count for task-specific intrinsic reward[J]. Advances in Neural Information Processing Systems, 2022, 35: 30432-30445.
>
> [2] Wan S, Tang Y, Tian Y, et al. DEIR: efficient and robust exploration through discriminative-model-based episodic intrinsic rewards[C]//Proceedings of the Thirty-Second International Joint Conference on Artificial Intelligence. 2023: 4289-4298.

---

> ### Author Response · Authors · 2024-08-14
> **DrQv2 Agent for MiniGrid Experiments**
>
> To further address your concerns, we have conducted additional experiments using DrQv2 as the underlying agent in the MiniGrid environment and compared the performance of the VCSE and SI2E exploration methods.
> The results indicates that the superior performance of our method is not limited to a specific agent but is broadly applicable and effective across a range of modern reinforcement learning models.
>
> $\bullet$ Success Rate：
> |MiniGrid|RedBlueDoors-$6$x$6$|SimpleCrossingS$9$N$1$|KeyCorridorS$3$R$1$|DoorKey-$6$x$6$|DoorKey-$8$x$8$|Unlock|
> |:-------:|:-------:|:-------:|:-------:|:-------:|:-------:|:-------:|
> |DrQv2|$31.27 \pm 14.58$|$90.23 \pm 10.25$|$78.62 \pm 5.39$|$87.49 \pm 7.21$|-|$93.29 \pm 9.46$|
> |DrQv2+VCSE|$82.05 \pm 9.17$|$92.13 \pm 6.73$|$83.27 \pm 2.33$|$94.27 \pm 2.15$|$90.15 \pm 23.71$|$93.13 \pm 5.77$|
> |DrQv2+SI2E|$89.71 \pm 4.93$|$95.27 \pm 3.00$|$90.07 \pm 0.57$|$96.47 \pm 1.71$|$97.21 \pm 4.49$|$98.33 \pm 2.96$|
>
> $\bullet$ Required Step:
> |MiniGrid|RedBlueDoors-$6$x$6$|SimpleCrossingS$9$N$1$|KeyCorridorS$3$R$1$|DoorKey-$6$x$6$|DoorKey-$8$x$8$|Unlock|
> |:-------:|:-------:|:-------:|:-------:|:-------:|:-------:|:-------:|
> |DrQv2|-|$614.71 \pm 34.72$|-|$439.23 \pm 109.24$|-|$538.22 \pm 97.36$|
> |DrQv2+VCSE|$1530.92 \pm 237.18$|$179.34 \pm 26.14$|$359.22 \pm 13.29$|$517.44 \pm 31.23$|$2103.49 \pm 197.33$|$514.00 \pm 72.93$|
> |DrQv2+SI2E|$847.06 \pm 74.39$|$134.97 \pm 19.77$|$198.71 \pm 6.31$|$251.17 \pm 14.94$|$1318.46 \pm 142.57$|$293.49 \pm 46.71$|

---

### Official Review · Reviewer_i1kZ · 2024-07-14

**Soundness:** 4
**Presentation:** 3
**Contribution:** 3
**Rating:** 7
**Confidence:** 4

**Summary:**

This paper proposes SI2E framework based on structural information principles to overcome inherent limitations in traditional information theory as applied to Reinforcement Learning (RL). The SI2E framework innovatively quantifies the dynamic uncertainties inherent in state-action transitions through a metric termed as 'structural entropy'. The authors define a novel concept, 'structural mutual information', to derive state-action representations that are particularly relevant to the dynamics of the environment. Additionally, they introduce a value-based approach to structural entropy, which is designed to enhance the agent's exploratory capabilities within the RL landscape. The paper establishes robust theoretical connections with established information-theoretic methodologies, thereby substantiating the rationality of the SI2E framework. Empirical validations through comparative experiments in the MiniGrid and DeepMind Control Suite environments underscore the framework's superiority over existing exploration benchmarks, showcasing its practical efficacy.

**Strengths:**

1. Innovative Contribution: The paper presents a novel framework, SI2E, which innovatively applies structural information principles to enhance reinforcement learning. This approach addresses key challenges in traditional information theory within the RL domain, offering a fresh perspective for exploration.
2. Clarity and Organization: The manuscript is exceptionally well-organized and written with exceptional clarity, making complex concepts easily comprehensible to readers.
3. Theoretical Rigor: The authors provide a solid theoretical foundation for their method, supported by comprehensive proofs that validate the framework's rationality and effectiveness.
4. Empirical Validation: The paper includes detailed experimental verifications across a variety of scenarios, which not only demonstrate the effectiveness of the SI2E framework but also highlight its robustness and superiority over current exploration methods.
5. Comprehensive Appendices: The paper's appendix is thorough, offering additional theoretical proofs, detailed model descriptions, and supplementary experimental data, which further substantiate the paper's claims.
6. Overall Impact: The paper demonstrates significant strength in both theoretical development and empirical validation, contributing valuable insights to the field of reinforcement learning and warranting acceptance for publication.

**Weaknesses:**

1. Although the paper includes a time complexity analysis, there is a gap in understanding the actual wall time required for training. Providing insights into the practical training duration would offer a more tangible assessment of the method's efficiency.
2. The current related work section provides a succinct overview. To strengthen the paper's context and background, a more detailed discussion of classical information-theoretic methodologies and structural information principles in the appendix would be beneficial.

**Questions:**

Please see the above weaknesses.

**Limitations:**

Yes

---

> ### Author Rebuttal · Authors · 2024-08-01
>
> We systematically address each of your queries, labeling weaknesses as 'W' and questions as 'Q'.
> Please note that, unless otherwise specified, any table or figure refers to the supplementary results in the Author Rebuttal PDF.
>
> $\bullet$ W1: Practical Training Time.
> We have comprehensively analyzed the training time for our method, SI2E, and two baselines, SE and VCSE.
> Specifically, we measure each method's time expenditure for a single training step as:
>     SI2E with $8.41 \pm 0.15$ ms,
>     SE with $8.60 \pm 0.23$ ms,
>     VCSE with $8.15 \pm 0.17$ ms.
> Our findings indicate that our method maintains comparable training time expenditure to the other methods, demonstrating its practicality.
> By providing these practical measurements, we underscore that SI2E is robust in theoretical time complexity and real-world training scenarios, ensuring efficiency and feasibility in practical applications.
>
> $\bullet$ W2: Related Work.
> We have reorganized the related work section in the appendix into three subsections: Maximum Entropy Exploration, Representation Learning, and Structural Information Principles.
>
> In this work, we leverage structural information principles to derive the hierarchical state-action structure and define value-conditional structural entropy as an intrinsic reward, leading to more effective agent exploration.
> Compared to current maximum entropy explorations, SI2E introduces and minimizes sub-community entropy to motivate the agent to explore specific sub-communities with high policy values.
> This approach avoids redundant explorations in low-value sub-communities and enhances maximum coverage exploration.
> Unlike the VCSE baseline, our method enables balanced exploration without needing prior knowledge of downstream tasks, effectively addressing previous approaches' limitations.
>
> Additionally, our work introduces structural mutual information to measure the structural similarity between two variables for the first time.
> We present an innovative embedding principle that incorporates the representation variable's entropy and mutual information with state variables, more effectively eliminating irrelevant information than the traditional Information Bottleneck principle.

---

### Author Rebuttal · Authors · 2024-08-06

We are immensely grateful for all reviewers' insightful comments, which have guided a comprehensive refinement of our manuscript.
Please note that supplementary experimental results, including two tables and three figures, are available in the PDF.
Unless otherwise specified, any table or figure refers to these supplementary results.

$\bullet$ **Related Work.**
We have reorganized the related work section in the appendix into three subsections: Maximum Entropy Exploration, Representation Learning, and Structural Information Principles.

In this work, we leverage structural information principles to derive the hierarchical state-action structure and define value-conditional structural entropy as an intrinsic reward, leading to more effective agent exploration.
Compared to current maximum entropy explorations, SI2E introduces and minimizes sub-community entropy to motivate the agent to explore specific sub-communities with high policy values.
This approach avoids redundant explorations in low-value sub-communities and enhances maximum coverage exploration.
Unlike the VCSE baseline, our method enables balanced exploration without needing prior knowledge of downstream tasks, effectively addressing previous approaches' limitations.

Additionally, our work introduces structural mutual information to measure the structural similarity between two variables for the first time.
We further present an innovative embedding principle that incorporates the representation variable's entropy and mutual information with state variables, more effectively eliminating irrelevant information than the traditional Information Bottleneck principle.

$\bullet$ **Generation of Optimal Encoding Trees.**
We have provided additional explanations and illustrative examples for the encoding tree optimization on $\mathcal{T}^2$.
As shown in Figure 2(a), the stretch operator is executed over sibling nodes $\alpha_i$ and $\alpha_j$ that share the same parent node, $\lambda$.
The detailed steps of this operation are as follows:
\begin{equation}
    {\alpha^\prime}^- = \lambda\text{,}\quad {\alpha_i}^-=\alpha^\prime\text{,}\quad {\alpha_j}^-=\alpha^\prime\text{,}
\end{equation}
where $\alpha^\prime$ is the added tree node via the stretch operation.
The corresponding variation in structural entropy, $\Delta H$, due to the stretch operation is calculated as follows:
\begin{equation}
    \Delta H = -\frac{g_{\alpha_i} + g_{\alpha_j} - g_{\alpha^\prime}}{\operatorname{vol}(G)} \cdot \log \frac{\operatorname{vol}(\alpha^\prime)}{\operatorname{vol}(G)} \text{.}
\end{equation}
At each iteration, the HCSE algorithm greedily selects the pair of sibling nodes that cause the maximum entropy variation, $\Delta H$ to execute one stretch optimization.

For $G_{sa}$, which involves only a single state-action representation variable, $Z_t$, we do not use the $2$-layer approximate binary tree constraint from SMI.
Instead, we exhaustively search all encoding trees that meet the height constraint to identify the optimal one.
Specifically, we apply the stretch and compress operators from the HCSE algorithm to iteratively and greedily optimize the encoding tree for $G_{sa}$.
A visual explanation is provided in Figure 2(b), which intuitively illustrates the optimization process.

$\bullet$ **Intuition Behind Figure 1.**
Figure 1 in our manuscript illustrates a six-state Markov Decision Process (MDP) with four actions. The different densities of the blue and red lines represent different actions, as indicated in the legend, resulting in state transitions with the objective of returning to the initial state $s_0$.
Solid lines specifically denote actions $a_0$ and $a_1$.
The transitions between states $s_2$ and $s_5$ are considered redundant because they do not contribute to the primary objective of returning to $s_0$.
Therefore, the state-action pairs $(s_2, a_0)$ and $(s_5, a_1)$ have lower policy values.

A policy maximizing state-action Shannon entropy would encompass all possible transitions (blue color).
In contrast, these redundant state-action pairs are grouped into a sub-community by leveraging structural information.
The policy based on value-conditional structural information minimizes the entropy of this sub-community to avoid visiting it unnecessarily.
Simultaneously, it maximizes state-action entropy, resulting in maximal coverage for transitions (red color) that are more likely to contribute to the desired outcome in the simplified five-state MDP.

In this scenario, at each timestep $t$, $s_t$, $s_{t+1}$, and $z_{t}$ denote the representations of the state before the transition, the state after the transition, and the corresponding state-action pair, respectively.

$\bullet$ **$2$-layer Approximate Binary Tree and $l$-transformation.**
Our approach utilizes $2$-layer approximate binary trees as the structural framework for measuring the structural similarity between two variables.
The $2$-layer binary tree represents a one-to-one matching structure between variables, which facilitates the calculation of the minimum required bits to determine accessible vertices via a single-step random walk, e.g., the joint structural entropy of two variables.
Using a $2$-layer binary tree ensures computational traceability.
More complex structures will increase the cost of increased computational complexity.

The $l$-transformation systematically traverses all potential one-to-one matchings between the variables, providing a comprehensive measure of their structural similarity.
By employing the $l$-transformation, we guarantee that the matchings are unique and non-redundant.
The $l$-transformation is integral to the formal definition of SMI.

This choice might introduce certain limitations; however, our primary goal was to establish a foundational framework.
We recognize the potential for further exploration of alternative structures for defining SMI in future research.

---

### Decision · Program_Chairs · 2024-09-25

**Decision:**

Accept (poster)

**Comment:**

this paper proposes a "Effective Exploration framework", based on information theoretic principles.
The experiment results show in general improvement comparison to the existing works.
Most of the reviewers' concerns have been satisfactory addressed by the authors. This paper brings
new ideas and might be of significance for the machine learning community. It is recommended for acceptance to NeurIPS 24.